# A Narrative Review on Manufacturing Methods Employed in the Production of Mesenchymal Stromal Cells for Knee Osteoarthritis Therapy

**DOI:** 10.3390/biomedicines13020509

**Published:** 2025-02-18

**Authors:** Rasmus Roost Aabling, Maria Rusan, Anaïs Marie Julie Møller, Naija Munk-Pedersen, Carsten Holm, Brian Elmengaard, Michael Pedersen, Bjarne Kuno Møller

**Affiliations:** 1Comparative Medicine Lab, SDCA-Steno Diabetes Center Aarhus, Department of Clinical Medicine, Aarhus University, Palle Juul-Jensens Boulevard 99 and 11, DK-8200 Aarhus, Denmark; 2Department of Molecular Medicine, Aarhus University Hospital, Brendstrupgårdsvej 21A, DK-8200 Aarhus, Denmark; marrus@rm.dk; 3Department of Clinical Pharmacology, Aarhus University Hospital, Palle Juul-Jensens Boulevard 99, DK-8200 Aarhus, Denmark; 4Department of Clinical Medicine, Aarhus University, Palle Juul-Jensens Boulevard 99, DK-8200 Aarhus, Denmark; bjamoell@rm.dk; 5Center for Gene and Cellular Therapy, Department of Clinical Immunology, Aarhus University Hospital, Palle Juul-Jensens Boulevard 99, DK-8200 Aarhus, Denmark; anaimo@rm.dk; 6Comparative Medicine Lab, Department of Clinical Medicine, Aarhus University, Palle Juul-Jensens Boulevard 99, DK-8200 Aarhus, Denmark; naijamunk@clin.au.dk (N.M.-P.); michael@clin.au.dk (M.P.); 7Department of Orthopedic Surgery, Elective Surgery Centre, Silkeborg Regional Hospital, Falkevej 1G, DK-8600 Silkeborg, Denmark; carsholm@rm.dk (C.H.); briaelme@rm.dk (B.E.)

**Keywords:** mesenchymal stromal cells, knee osteoarthritis, cell culture, cryopreservation, cell storage, thawing, reconstitution, viability, advanced therapy medicinal product

## Abstract

Knee osteoarthritis (OA) is a chronic, progressive, inflammatory, and degenerative whole-joint disease. Early-stage OA treatments typically include physiotherapy, weight-loss, pain relief medications, and intra-articular knee injections, such as corticosteroids, hyaluronic acid, or platelet-rich plasma. These treatments primarily provide symptomatic relief rather than reversing or halting disease progression. Recently, mesenchymal stromal cell (MSC) injections have garnered attention due to their immunomodulatory and regenerative capacities. MSCs, which can be derived from sources such as bone marrow, umbilical cord, or adipose tissue, and can be allogeneic or autologous, have demonstrated promising results in both animal models and several human studies. However, different protocols have been employed, presenting challenges for comparing outcomes. In this review, we address these variable settings, evaluate current practices, and identify key factors critical in optimizing MSC-based therapies by critically reviewing clinical trials of ex vivo expanded MSC therapies for OA undertaken between 2008 and 2023. Specific attention was given to two key aspects: (1) the cell culture process employed in manufacturing of autologous or allogeneic MSC products, and (2) the post-culture methods employed in storage, reconstitution and administration of the MSCs. Our findings suggest that standardizing MSC production for clinical applications remains a significant challenge, primarily due to variations in tissue sources, harvesting techniques, and manufacturing protocols, and due to broad discrepancies in reporting. Thus, we propose a set of minimal reporting criteria to guide future clinical trials. A common reporting guideline is a critical step towards a more standardized MSC production across different laboratories and clinical settings, thereby enhancing reproducibility and advancing the field of regenerative medicine for knee OA, as well as other disease settings.

## 1. Introduction

Knee osteoarthritis (OA) is a chronic, progressive, inflammatory, degenerative disease [1], resulting in disabling local pain and stiffness. OA also leads to decreased knee functionality and eventually articular deformities [2]. OA is a whole-joint disease that involves cartilage and meniscal degeneration and inflammation, as well as fibrosis of the infrapatellar fat pad and the synovial membrane [3]. On X-rays, some of these features can be observed as joint space narrowing (a sign of degeneration of the articular knee cartilage), osteophyte formation, subchondral sclerosis and bony deformities [4].

Current estimates indicate that OA affects approximately 4% of the world’s population, translating to around 250 million individuals [5]. In adults aged 55 years and older, the prevalence of OA rises to 13%, with this figure escalating with advancing age [6]. The substantial disease burden and high prevalence of OA contribute to considerable healthcare costs. In the United States of America, OA generates over USD 27 billion in annual healthcare expenditures [7], a figure projected to rise in tandem with the rapidly aging demographic [8,9,10].

Early-stage treatment options for OA are primarily focused on symptom relief and typically involve physiotherapy, weight-loss, pain-relieving medication, and, in certain cases, intra-articular knee injections.

As OA progresses and the knee sustains severe degeneration, total knee replacement remains the only standard treatment option [11].

Common intra-articular injections often comprise corticosteroids. However, there is evidence that these corticosteroids may damage the articular cartilage of the knee, thereby accelerating the degenerative process [12,13]. Another option for intra-articular therapy is hyaluronic acid (HA), a naturally occurring glycosaminoglycan that offers viscous lubrication and shock-absorbing properties and may have anti-inflammatory effects [14]. Clinical trials investigating HA injections have, however, produced inconsistent results regarding their efficacy in pain alleviation [14]. In recent years, platelet-rich plasma (PRP) injections prepared from an autologous blood sample have emerged as an additional treatment option for OA. PRP therapy draws on coagulation factors, cytokines and various platelet proteins to reduce the ongoing inflammatory OA process. Studies have shown significant pain relief after 2 months, with effects lasting as long as 12 months. Although promising, PRP therapy still lacks strong evidence to substantiate its clinical application, as well as standardization of preparation, dosages, and frequency of administration [14]. These treatment strategies, focusing largely on symptomatic management, are often insufficient, highlighting a critical need for therapeutic strategies that not only alleviate pain and enhance functionality but also effectively reverse or halt the progression of the disease.

Advanced therapy medicinal products (ATMPs), such as mesenchymal stromal cells (MSCs), also called mesenchymal stem cells, have recently received considerable interest as a novel treatment for degenerative diseases, including OA. This interest in MSCs is not only due to their ability to differentiate into mesenchymal cells residing in bone, cartilage and adipose tissues [15], but also due to their stimulatory effects on endogenous host cells. MSCs have been shown to enhance the secretion of growth factors, such as transforming growth factor-beta (TGF-β), bone morphogenetic proteins (BMP), and connective tissue growth factor (CTGF), that promote tissue regeneration [16,17]. MSCs also have the capacity to inhibit the expression of inflammatory cytokines while promoting the secretion of anti-inflammatory factors, such as interleukin(IL)-10, interleukin-1 receptor antagonist IL1-RA, and IL-13 [16].

In OA animal models, intra-articular injection of MSCs is safe and able to ameliorate symptoms and signs associated with knee OA and may have the potential to regenerate lost knee cartilage [18,19,20,21,22]. A recent meta-analysis by Song et al. (2020), based on 19 clinical trials, further corroborated these findings, concluding that intra-articular knee administration of MSCs is safe, with minimal risk of adverse events [23]. This meta-analysis also indicated that MSCs can significantly reduce pain and enhance functionality. Today, there are sparse results regarding cartilage regeneration with MSC treatments in humans, and less than a handful of clinical trials have been able to demonstrate cartilage thickening [24,25,26]. The majority of trials show no difference from baseline cartilage values, suggesting cessation of the disease process rather than reversal [27,28,29,30].

MSCs for clinical applications can be isolated from various tissues, including bone marrow, umbilical cord, synovial membrane, muscle or adipose tissue (i.e., subcutaneous fat or infrapatellar fat pad) [31,32]. These MSCs may be utilized in a one-step procedure directly in the operating room, such as microfragmented adipose tissue and concentrated bone marrow aspirates, or they can undergo further isolation and expansion in a laboratory setting. One-step separation processes often yield low cellular concentrations, of typically <1 × 10^6^ MSCs/mL [31,33,34,35]. MSC isolation and in vitro culture expansion are generally necessary to produce higher concentrations (1–200 × 10^6^ MSCs/mL), which some argue are essential for clinical improvement and effective tissue regeneration [36].

The majority of clinical trials exploring MSC therapy for OA have predominantly utilized autologous transplants, relying on freshly harvested MSCs [36]. However, this method presents logistical challenges, necessitating meticulous coordination for MSC procurement, operating room availability, and timely treatment scheduling. An alternative strategy involves the use of off-the-shelf pre-expanded cryopreserved allogeneic MSCs, which alleviates these practical difficulties. Allogeneic MSC therapies also present a cost-effective approach, enabling the treatment of multiple patients using material from one donor, and securing a more standardized product.

Nevertheless, concerns have been raised regarding the potential cytotoxic effects of cryoprotectants, such as dimethylsulfoxide (DMSO) [37]. Hence, for localized treatments, it may be preferable to wash and reconstitute the MSC product following cryopreservation to remove cryoprotectants. This approach helps reduce potential cytotoxicity at the administration site, which is particularly important for poorly vascularized recipient sites, such as in intra-articular areas.

This narrative review aims to provide a comprehensive overview of the current state of MSC therapy for knee OA, focusing on two primary aspects: (1) the cell culture process utilized in manufacturing of autologous or allogeneic MSC products expanded ex vivo, and (2) the post-culture methods employed for storage, reconstitution and administration of MSCs. We seek to describe the efficacy and safety of MSC therapies and to propose a set of minimal reporting criteria to facilitate interpretation of clinical trial results, improve standardization and reproducibility in the field, and advance regenerative medicine.

### 1.1. Manufacturing of MSCs

Manufacturing MSCs encompasses a variety of processing steps before being administered to the patient (Figure 1). These steps are individually discussed, presenting a diversity of methodologies employed across clinical trials. Table 1 shows 33 OA clinical trials with MSCs, Table 2 shows the manufacturing processes of MSCs in the included studies, Table 3 shows the post-culture and reconstitution of MSCs and Table 4 shows the quality control measures during MSC manufacturing. Table 5 presents minimal reporting criteria for future clinical trials. An extended version of Table 1 is provided as Appendix A, including the radiographic Kellgren Lawrence grade of included patients, outcome measures, adverse events and clinical efficacies.

### 1.2. Cellular Product Characteristics and Culture Process

#### 1.2.1. Tissue Procurement

##### Tissue Source

MSCs employed in OA clinical trials have been derived from well-known and widely utilized mesenchymal tissue sources, such as the bone marrow, adipose tissue, or umbilical cord. In early studies, the bone marrow was the most commonly utilized MSC source, and the tissue procurement method described in these early studies remains the primary one used today [66]. The skin of the posterior iliac crest is locally anesthetized with aseptic technique, and the bone marrow is harvested. This is performed by puncturing the iliac bone with a trocar and aspirating the bone marrow through it. Multiple punctures are often required to obtain sufficient material for subsequent isolation and cultivation.

As the bone marrow isolation process is associated with pain and the MSC content is higher in other mesenchymal tissues, subsequent clinical studies shifted towards utilizing MSCs harvested from adipose tissue (adipose-derived MSCs, AD-MSCs) and from the umbilical cord (umbilical cord-derived MSCs, UC-MSCs) [31].

OA therapy with AD-MSCs was introduced in 2014 [58] and employed in several studies in subsequent years [27,50,51,53].

The procurement process was comprehensively explained in the randomized control trial by Lee et al. (2019) [29]. Briefly, adipose tissue is harvested through liposuction, typically from the abdominal subcutaneous fat. Using aseptic technique, the skin is prepared and locally anesthetized. A small 2 mm stab incision of the skin is performed, and the subcutaneous fat layer is infiltrated with a tumescent solution usually consisting of lidocaine, adrenalin and bicarbonate suspended in a saline solution [28]. After waiting 10–15 min for the tumescent solution to take effect, a liposuction cannula is introduced through the stab incision to harvest the adipose tissue.

Building on successful safety and feasibility studies with allogeneic transplantation of both bone marrow-derived MSCs (BM-MSCs) and AD-MSCs, the first patient with knee OA was treated with UC-MSCs in 2018 [47]. Out of 33 clinical trials identified, only five trials conducted intra-articular knee injection with UC-MSCs [25,39,40,43,47]. In these trials, umbilical cords were collected from full-term pregnancies of human placentas. The cords were washed, and the Wharton’s jelly tissue was dissected from the umbilical vein and artery. The tissue was then cut or minced and placed in a culture flask to undergo further ex vivo culturing from the explant.

*Summary of paragraph:* Several clinical trials have been conducted with MSCs. During the first years of OA MSC therapy, bone marrow was used as a cellular source. In the following years, the source shifted to adipose tissue or umbilical cord given the higher cellular content. The harvest procedure depends on the tissue source and clinical trials have utilized similar approaches.

#### 1.2.2. Cell Isolation 

After tissue harvesting, the material is transported to the manufacturing facility. A clinical MSC production laboratory necessitates an A-in-B laboratory setting and adherence to current Good Manufacturing Practice (GMP) to produce MSCs with sufficiently high sterility.

The cell isolation process differs across tissue of origin but is quite similar within tissue of origin across the trials considered here, as outlined below.

##### BM-MSC

The extracted bone marrow is placed in cell culture medium and nucleated cells are isolated by density gradient centrifugation. Isolated cells are then washed and further cultured.

##### AD-MSC

To isolate cells from the extracted adipose tissue, collagenase is added, and the suspension is digested for 45–60 min at 37 °C with intermittent shaking. The digestion process is stopped by adding cell culture medium and the cells are centrifuged. The supernatant is then discarded, and the pellet is reconstituted. This stromal vascular fraction (SVF) is filtered and further cultivated.

##### UC-MSC

Four out of five studies cultured isolated Wharton’s jelly tissue by the “outgrowth from explant” method. Whole pieces of Wharton’s jelly tissue were placed in culture flasks and UC-MSCs were grown from this tissue onto the plastic bottom of the culture flasks. Soltani et al.’s (2019) is the only study to employ a different method, subjecting minced Wharton’s jelly tissue to further isolation [25]. After washing three times in a 9% sodium chloride solution, collagenase was added and the tissue was left for digestion for 3 h at 37 °C with intermittent shaking every 30 min. The 9% sodium chloride solution was added again, and the tissue solution was shaken, centrifuged, and then cultured.

#### 1.2.3. Expansion of MSC

##### Cell Counting

Prior to cell culture, cell counting was performed either by flow-cytometry or trypan blue cell staining and direct counting in Bürker-Türk Counting chambers. In total, 9 out of the 33 trials performed an estimated counting of the cells before cultivation and reported an estimated cell cultivation density per cm^2^ that ranged between 0.16 × 10^6^ cells/cm^2^ [24] and 1 × 10^6^ cells/cm^2^ [66].

##### Cell Culture

Cell culture medium ingredients differed between the clinical trials, but the majority of trials used a combination of Minimum essential medium (MEM) alpha or Dulbecco’s Modified Eagle Medium (DMEM) with antibiotics and a growth supplement. The majority of early clinical trials employed fetal bovine serum (FBS) as a growth supplement. However, concerns were raised that employing animal-derived growth supplements may lead to clinical complications including exposure to animal-derived (xeno) antigens and transmission of infectious agents. Thus, various pre-clinical ex vivo studies were conducted to evaluate the use of xeno-free growth supplements. These revealed that FBS and xeno-free growth supplements provided comparable proliferation rate, morphology and trilineage differentiation potential along with similar immunophenotypes [67,68]. These results have been translated to clinical trials, with nine of the trials considered herein using human serum (platelet lysate (PL) or PRP) as a growth supplement.

More than half of the clinical trials noted their incubation settings, with most utilizing standard growth conditions, specifically, a humidified incubator at 37 °C with 5% CO_2_. Only one trial, conducted by Freitag et al. (2019), employed hypoxic conditions, although the rationale for this decision was not specified [28]. The rationale for employing hypoxic conditions in the cultivation of MSCs may stem from the natural environment in which these cells typically reside. Additionally, previous research has shown that hypoxic conditions can significantly enhance growth kinetics, maintain genetic stability, and upregulate the expression of chemokine receptors during the in vitro expansion of MSCs [69].

Most trials changed the culture medium and removed non-adherent cells after 24–72 h (or 5–7 days in some studies [25,48,51,54,62]). Cells were cultured until they reached a 70–90% confluency and were then detached from their culture flasks by trypsinization for further subculture until the desired cell number was reached. Between passaging, the culture medium was changed every 3–6 days. The majority of clinical trials passaged 4–5 times. Only one RCT, by Soltani et al. (2019), cultured their UC-MSCs to passage 12 [25]. Research has consistently demonstrated a strong correlation between cellular senescence and prolonged passaging, especially when the number of passages exceeds 6 to 9, underscoring the importance of minimizing the number of passages to maintain cellular vitality and functionality [70,71].

*Summary of paragraph:* Cultivation of MSCs is usually undertaken using a growth medium, such as MEM or DMEM with antibiotics and a growth supplement. FBS has been used as a growth supplement, but more recently, xeno-free growth supplements (such as PL or PRP) are utilized to reduce risk of exposure to xeno-antigens and transmission of infectious agents. Standard incubator settings are usually a humidified incubator at 37 °C and 5% CO_2_. Initial cultivation is typically 24–72 h, following changing of the cultivation medium, and removal of non-plastic adherent cells. MSCs are usually grown to 70–90% confluence before they are detached from the culture flask by trypsinization for further subculture. MSCs are generally cultured for 4–5 passages.

#### 1.2.4. Cell Harvest

When the desired cell count has been achieved, the cells are detached from the cultivation flasks with trypsinization using trypsin-EDTA [52], TrypLE select [55] or TrypLE express [25], and the cells are then formulated in the final vehicle medium or prepared for cryo-storage.

#### 1.2.5. Storage

##### Freshly Harvested MSCs

The majority of the clinical trials included (20 out of 33) utilized freshly harvested cells following expansion. Consequently, these cells were formulated directly into the desired final vehicle medium following harvesting and washing of the cells, eliminating the need for cryopreservation and storage.

##### Cryopreserved MSCs

Out of the 13 clinical trials that used cryopreserved cells, four trials [39,41,47,65] used a combination of both cryopreserving, thawing, re-cultivation and usage of freshly harvested cells. The remaining nine trials reconstituted and administered the cells.

Centeno et al. (2011) used one such combination [72], cultivating BM-MSCs to ≈80% confluence, harvesting, reconstituting in either PL or platelet poor plasma, and cryopreserving at −80 °C using a controlled-rate freezing device. Rate-controlled freezing is a system specifically designed to achieve a rate of cooling of 1 °C per minute. This can either be accomplished by using a CoolCell (Corning, NY, USA) container or an automated controlled-rate freezer.

After 24–72 h, the cells were moved either to a −150 °C freezer or liquid nitrogen for long-term storage. Gradual freezing at a controlled slow rate, initially at −80 °C and subsequently transferred to cooler storage one or two days later, remains widely employed in the field. Failing to utilize this approach may result in the formation of ice crystals inside the cells, leading to increased cellular osmolality and subsequent dehydration, which can cause cell damage and death. Therefore, it is critical that the cooling rate during freezing is sufficiently slow to allow an appropriate volume of water to exit the cells [73].

Aside from that of Centeno et al. (2011), only a limited number of clinical trials have provided a detailed explanation of their freezing procedures. Gupta et al. (2016) utilized a widely accepted method of cryopreservation that involved addition of DMSO to the cells followed by gradual freezing [52]. The incorporation of membrane-permeable cryoprotectants such as DMSO or glycerol, in conjunction with gradual freezing, is currently regarded as the gold standard for clinical applications [74].

The addition of cryoprotectants is essential to the cryopreservation process as it inhibits the formation of intracellular ice crystals, thereby preventing cell death [37].

Unfortunately, MSC exposure to DMSO can result in cytotoxicity [37]. Francois et al. (2012) found that cryopreserving and subsequently thawing brain-derived MSCs from two different donors in an alpha MEM medium containing 30% FBS and 5% DMSO resulted in cellular viabilities of 61% and 41% compared to freshly cultured MSCs, which exhibited viabilities of 92% and 91% [75].

In parallel, Pal et al. (2012) found that embryonic stromal cells subjected to three different doses of DMSO (0.01% low dose, 0.1% medium dose, 1.0% high dose) showed substantially altered cell morphology and attachment potential. Cell viability was also significantly reduced, in a dose-dependent manner [76]. Thus, research has aimed to reduce the DMSO amount or to combine it with other cryoprotective agents for successful long-term storage.

Liu et al. (2010) found that cryopreserving BM-MSCs in 7.5% DMSO, 2.5% polyethylene glycol (PEG), and 2% bovine serum albumin (BSA) resulted in cell viabilities comparable to those obtained from BM-MSCs preserved in 10% DMSO [77]. The usage of animal components can, however, be problematic, and a xeno-free alternative for cryopreservation has therefore been developed [37]. In the work of Park et al. (2018), the effect of cryopreservation of AD-MSCs in different xeno-free compounds compared to FBS was investigated (10% DMSO and 0.9% NaCl, and either human serum albumin (HSA), human serum or serum free growth media (knockout serum replacement)). They demonstrated that the AD-MSC properties, including growth, immunophenotypes, gene expression patterns, and the trilineage differentiation potential, were similar to FBS for all compounds tested [78]. Recently, the combination of a xeno-free and DMSO-free cryo-agent has emerged [45].

In four clinical trials, a combination of first cryopreserving and later expanding and freshly harvesting the cells was utilized for knee OA treatment. After Centeno et al. presented this particular cultivation process, Matas et al. used it to cultivate UC-MSCs to P3, cryopreserved them, and then, prior to therapy, thawed the cells and further cultivated them to P5. The cells were then detached and washed twice with PBS and suspended in the final vehicle medium [47]. The same process was employed in two more recent publications, where cryopreserved cells were thawed 3–5 days before the intra-articular transplantation and cultivated to be used fresh [39,41].

*Summary of paragraph:* After cultivation, freshly harvested MSCs are usually washed and reconstituted in the final vehicle medium without any storage.

Only a few of the clinical trials utilized MSCs that had been cryopreserved, thawed, further cultivated and used freshly harvested. 

The gold standard for cryopreserving MSCs involves adding a permeable cryoprotectant, such as DMSO, to the cellular suspension immediately after harvest and then gradually freezing the cells. However, DMSO is known to be cytotoxic, leading to the emergence of DMSO-free cryoprotective agents in recent years.
biomedicines-13-00509-t003_Table 3Table 3Post-culture and reconstitution of MSCs in the clinical trials.No.AuthorsMSC Type Transplantation TypeFresh/CryoThawing Medium and Procedure Final Vehicle Medium Final Vehicle Amount Total MSCMSC Concentration (cells/mL)Cannula Gauge (G) Size for IAIIAI Administration Procedure1Kim et al. [38]AD-MSCAutoCryoNS2.1 mL of normal saline and 0.9 mL auto serum3 mL100 × 10^6^33.3 × 10^6^NSGuided by ultrasound by a specialized physician2Günay et al. [39]UC-MSCAlloCryo followed by thawing and 5 days of culturing before injection and used fresh5 days before intra-articular injection, cells were thawed and re-platedPhysiologic isotonic solution5 mL100 × 10^6^20 × 10^6^20GBlinded3Samara et al. [40]UC-MSCAlloCryoNSNSNSFirst injection: 34–50 × 10^6^, Second injection: 33–55 × 10^6^NSNSGuided by ultrasound by a senior interventional radiologist4Kim et al. [30][38]




5Jianrui et al. [41]BM-MSCAutoCryo followed by thawing and 3 days of cell culture before injection and used freshNSAuto PL3 mLNSNSNSBlinded6Lu et al. [42]AD-MSCAlloFreshNAElectrolyte solution containing 3.5% (*w*/*v*) amino acids and 7% (*w*/*v*) vitamins3 mL(1) low-dose group (10 × 10^6^), (2) mid-dose group (20 × 10^6^), (3) high-dose group (50 × 10^6^)(1) low-dose group (3.3 × 10^6^), (2) mid-dose group (6.7 × 10^6^), (3) high-dose group (16.7 × 10^6^)NSBlinded7Dilogo et al. [43]UC-MSCAlloFreshNA2 mL secretome and 2 mL HA4 mL10 × 10^6^2.5 × 10^6^NSNS8Chahal et al. [44]BM-MSCAutoFreshNA6.5 + 1.5 mL (2.5% of the patient’s auto serum in Plasmalyte A)8 mL(1) low-dose group (1 × 10^6^), (2) mid-dose group (10 × 10^6^), (3) high-dose group (50 × 10^6^), (4) mixed-dose group (1, 10, 50 × 10^6^)(1) low-dose group (0.125 × 10^6^), (2) mid-dose group (1.25 × 10^6^), (3) high-dose group (6.25 × 10^6^), (4) mixed-dose group (0.125, 1.25, 6.25 × 10^6^)NSGuided by ultrasound 9Lu et al. [26]AD-MSCAutoFreshNAElectrolyte solution containing 3.5% (*w*/*v*) amino acids and 7% (*w*/*v*) vitaminsApprox. 2.5 mL50 × 10^6^20 × 10^6^NSBlinded10Lee et al. [29] AD-MSCAutoCryoNSSaline3 mL100 × 10^6^33.3 × 10^6^NSGuided by ultrasound by a specialized physician11Freitag et al. [28]AD-MSCAutoCryoThawed, centrifuged and washed in chilled PBSSterile isotonic (0.9%) normal saline3 mL100 × 10^6^33.3 × 10^6^NSGuided by ultrasound 12Yokota et al. [45]AD-MSCAutoCryoRapidly thawed then cleansed in lactated Ringer’s solution. Immediately before direct injectionRinger’s solution NS12.8 × 10^6^ ± 2.9 × 10^6^NS23GBlinded. Injection performed by two senior orthopedic surgeons 13Zhao et al. [46]AD-MSCAlloCryoNSNSNS(1) low-dose group (10 × 10^6^ cells), (2) mid-dose (20 × 10^6^) (3) high-dose group (50 × 10^6^)NSNSNS14Soltani et al. [25]UC-MSCAlloFreshNANS10 mL50–60 × 10^6^5–6 × 10^6^NSBlinded15Bastos et al. [34] Same as [48]



16Matas et al. [47]UC-MSCAlloCryo followed by thawing and passaging until P5, then detached and washed twice with PBS and suspended in final vehicle mediumNSSaline solution and 5% AB plasma3 mL20 × 10^6^6.6 × 10^6^NSBlinded. Injection performed by two senior orthopedic surgeons17Bastos et al. [48]BM-MSCAutoFreshNAPBS supplemented with 2% HSA 10 mL40 × 10^6^4 × 10^6^20GBlinded18Emadedin et al. [49]BM-MSCAutoFreshNASaline supplemented with 2% HSA5 mL40 × 10^6^8 × 10^6^22GBlinded19Song et al. [50]AD-MSCAutoFreshNAElectrolyte solution containing 3.5% (*w*/*v*) amino acids and 7% (*w*/*v*) vitamins3 mL(1) low-dose group (10 × 10^6^), (2) mid-dose group (20 × 10^6^), (3) high-dose group (50 × 10^6^)(1) low-dose group (3.3 × 10^6^), (2) mid-dose group (6.6 × 10^6^), (3) high-dose group (16.7 × 10^6^)NSGuided by ultrasound 20Spasovski et al. [51]AD-MSCAutoFreshNAPBS 1 mL5–10 × 10^6^5–10 × 10^6^NSBlinded21Kuah et al. [27]AD-MSCAlloCryoThawed and drawn straight into a syringe and IAICell culture media and cryopreservative2 mL(1) low-dose group (3.9 × 10^6^), (2) high-dose group (6.7 × 10^6^)(1) low-dose group (1.95 × 10^6^), (2) high-dose group (3.35 × 10^6^)NSGuided by ultrasound by a radiologist or sports and exercise medicine physician22Al-Najar el al [24]BM-MSCAutoFresh NA0.9% normal saline5 mL30.5 × 10^6^6.1 × 10^6^NSBlinded. Performed by an experienced orthopedic surgeon23Gupta et al. [52]BM-MSCAlloCryoCryobags containing 200 × 10^6^ BM-MSCs with 15 mL of PLASMA-LYTE A (5% HSA and 10% DMSO) were thawed and diluted to 100 mLusing PLASMALYTE A. BM-MSC were centrifuged, and the cell pellet was resuspended in the final vehicle mediumPLASMA-LYTE A (5% HSA)2 mL: (1) dose level I group, (2) dose level II group 4 mL: (3) dose level III group, (4) dose level IV group(1) dose level I group (25 × 10^6^), (2) dose level II group (50 × 10^6^), (3) dose level III group (75 × 10^6^), (4) dose level IV group (150 × 10^6^)(1) dose level I group (12.5 × 10^6^), (2) dose level II group (25 × 10^6^), (3) dose level III group (37.5 × 10^6^), (4) dose level IV group (75 × 10^6^)20GBlinded by either an orthopedic surgeon or rheumatologist24Pers et al. [53]AD-MSCAutoFreshNA3.6% HSA and a polyionic solution containing glucose5 mL(1) low-dose group (2.0 × 10^6^), (2) mid-dose group (10 × 10^6^), (3) high-dose group (50 × 10^6^)(1) low-dose group (0.4 × 10^6^), (2) mid-dose group (2 × 10^6^), (3) high-dose group (10 × 10^6^)NSGuided by ultrasound25Soler et al. [54]BM-MSCAutoFreshNASaline solution supplemented with 2% HAS10 mL40.9 × 10^6^4.9 × 10^6^NSBlinded26Lamo-Espinosa et al. [55,56]BM-MSCAutoFresh NARinger’s lactate buffer containing 1 % HSA1.5 mL: (1) low-dose group, 3 mL: (2) high-dose group(1) low-dose group (10 × 10^6^), (2) high-dose group (100 × 10^6^)(1) low-dose group (6.7 × 10^6^), (2) high-dose group (33.3 × 10^6^)19GBlinded by three different orthopedic surgeons27Vega et al. [57]BM-MSCAlloFresh NARinger lactate solution containing 0.5% HSA and 5 mM glucose8 mL40 × 10^6^5 × 10^6^NSBlinded28Jo et al. [58,59]AD-MSCAutoNot explicitly stated (possibly fresh)NASaline 3 mL(1) low-dose group (10 × 10^6^), (2) mid-dose group (50 × 10^6^), (3) high-dose group (100 × 10^6^)(1) low-dose group (3.3 × 10^6^), (2) mid-dose group (16.6 × 10^6^), (3) high-dose group (33.3 × 10^6^)22GBlinded by an orthopedic surgeon29Orozco et al. [60,61]BM-MSCAutoFresh NARinger’s lactate solution containing 0.5% HSA and 5 mM glucose8 mL40 × 10^6^5 × 10^6^NSBlinded30Emadedin et al. [62]BM-MSCAutoFresh NAPhysiological serum4–5 mL20–24 × 10^6^5 × 10^6^NSGuided by fluoroscopy31Davatchi et al. [63,64]BM-MSCAutoFresh NANS5.5 mL8–9 × 10^6^1.6–1.45 × 10^6^NSBlinded32Centeno et al. [65]BM-MSCAutoFresh/Cryo Rapidly thawed and then transferred into warmed auto PL/Alpha-MEM and cultured for another passage before being used fresh**Fresh:** 10–20% Auto PL or Conditioned serum (A-PRP)—consisting of auto platelet rich plasma (CaCL_2_ (2.86–14.3 mg/mL) with human Thrombin (28.6–142.8 IU/)mL which had been incubated for 1 h to 6 days of which the platelets were then pelleted by centrifugation and the supernatant drawn off for re-injection with BM-MSCs. **Cryo:** 90–95% auto PL or in PRP. **NOTE:** BM-MSCs and one of the above-mentioned vehicle mediums were also injected together with contrast (Omnipaque 300 mg/mL-NDC; diluted 1:1 or 1:2 with PBS).NSFresh: NS/Cryo: 1–3 × 10^6^NSNSGuided by fluoroscopy33Centeno et al. [66]BM-MSCAutoFresh NAPBSNS22.4 × 10^6^NS25GGuided by fluoroscopyAD-MSC = Adipose-derived mesenchymal stromal cell, Allo = Allogeneic, Alpha-MEM = Alpha-Minimum essential medium, Auto = Autologous, BM-MSC = Bone marrow-derived mesenchymal stromal cell, Cryo = Cryopreserved, DMSO = Dimethyl sulfoxide, HA = Hyaluronic acid, HSA = Human serum albumin, IAI = Intra-articular injection, MSC = Mesenchymal stromal cell, NA = Not applicable, NS = Not specified, PBS = Phosphate-buffered saline, PL = Platelet lysate, UC-MSC = Umbilical cord-derived mesenchymal stromal cell.


### 1.3. Pre-Injection Reconstitution

A primary focus for MSC production facilities has been to investigate the impact of MSC thawing and reconstitution conditions, and it has been observed that variations in MSC handling can significantly affect both the recovery and viability of MSCs [79,80,81].

#### 1.3.1. Usage of Freshly Harvested MSC

The majority of the included clinical trials utilized freshly harvested MSCs. In this process, after reaching the desired cell count, the cells were harvested by trypsinization, washed and formulated in the final vehicle medium.

Several preclinical ex vivo studies evaluated a wide array of reconstitution solutions and conditions for various MSC types to identify the combination that would secure high cellular viability and stability.

In a study by Chen et al. (2013), the cell viability of UC-MSCs resuspended in either 0.9% saline, 5% dextrose, dextrose and sodium chloride, Plasma-Lyte A, 1% HSA, or 5% HSA were compared. The cells were stored in these solutions for 2, 4, or 6 h at 4 °C or room temperature (RT) (24 °C). After 6 h at 4 °C, cells formulated in saline, dextrose and Plasma-Lyte A had viabilities ranging between 80% and 95%. For the other solutions, cellular viability decreased to 80% and below after only 2 h of storage, with sodium chloride performing the worst. This deterioration came earlier for cells stored at RT, and following 6 h, viability was 80% or below for all solutions tested [82]. Furthermore, Gálvez-Martín et al. (2013) studied the effects of longer storage time in various solutions on freshly harvested AD-MSCs. Cellular viability was tested over 60 h at 8 °C. The solutions tested were Lactated Ringer’s solution + glucose 5% + Albumin 20% (Medium 1), Lactated Ringer’s solution + glucose 5% (Medium 2), Lactated Ringer’s solution (Medium 3) or Sodium Chloride 0.9% (Medium 4). Cells resuspended in Medium 1, 2 and 3 performed the best and upheld a viability >80% for 48, 36 and 30 h, respectively. After 12 h, cells in Medium 4 had viabilities of 75%, ranking as the lowest among the tested conditions [83]. Consequently, these preclinical ex vivo studies concluded that freshly harvested MSCs could be stored for several hours in balanced isotonic saline solutions at chilled temperatures.

The majority of clinical trials explained their pre-injection reconstitution methods in detail but their reporting about cell viability before administration generally lacked detailed information. Some trials conducted testing immediately before administration [44,51,53,57], while others performed testing and then shipped their product for later use [54,58]. This raises questions about the validity of viability percentages obtained hours prior to administration.

Among the clinical trials employing freshly harvested cells, Jo et al. (saline alone as the final vehicle medium) and Soler et al. (saline + HSA) reported viabilities >95% and 85%, respectively. Two other trials resuspended freshly harvested cells in Ringer’s lactate solution containing 0.5% HSA and 5 mM glucose [57,60] and another trial used Ringer’s lactate solution containing 1% HSA [55]. Before administration, their cellular viability ranged from 91% to >98%. Two clinical trials used Plasma-Lyte A, a balanced salt solution used for infusion purposes. The solution consists of the following substances in sterile water: sodium chloride, potassium chloride, magnesium chloride hexahydrate, sodium acetate trihydrate and sodium gluconate. Cell viability for one of the groups was similar to that of other trials at >85% [52], but for Chahal et al. (2019) only a viability of >70% was noted. The latter is the lowest viability reported in any of the trials employing freshly harvested MSCs, and lower than that reported in the mentioned ex vivo preclinical studies. The low viability may be related to the storage of cells at 15–25 °C during the first 8 h in Plasma-Lyte A [44].

In addition, a number of studies reconstituted MSCs in an electrolyte solution containing 3.5% (*w*/*v*) amino acids and 7% (*w*/*v*) vitamins [26,42,50]. None of these trials reported cell viability prior to administration.

#### 1.3.2. Usage of Cryopreserved MSC

Cryopreservation is necessary in off-the-shelf MSC therapy. On the day of injection, the cells are thawed, washed and reconstituted in a final vehicle medium to the desired concentration. In work conducted by our group, we optimized the post-cryopreservation process and found that even small adjustments in the MSC processing have notable effects on the stability and viability of the AD-MSCs. In particular, we found that the presence of protein or serum in the thawing medium was crucial, as protein and serum-free thawing solutions resulted in 50% cell loss and significantly decreased cell viability [84].

We performed viability analysis of AD-MSCs resuspended and stored for up to 4 h in either culture medium, Ringer’s acetate, saline, PBS or saline-HSA. Cells were stored at RT or wet ice. Despite the storage medium, after 4 h, viabilities of 85–95% could be observed when cells were kept on ice. When kept at RT, cells in both culture medium and PBS exhibited a significant decrease in cell viability, with a drop from >90% to <75% noticeable within one hour [84].

Similar results were obtained by Pal et al. (2008), comparing the cellular viability between freshly harvested and cryopreserved BM-MSCs, and they found cell viabilities of 90% after 4 h of storage at 4 °C using isotonic saline as the final vehicle medium. However, the findings by Pal et al. were achieved through a thawing method devoid of serum or protein additions [79], which, in our experimentation, led to a notable decline in viability and cell retrieval.

In several clinical trials, there is a lack of information about the thawing protocols, although the final vehicle medium is described [29,30,38,39,40,41,46,47].

In addition, as with the trials utilizing freshly harvested cells, a lack of viability reporting was observed with the clinical trials employing cryopreservation. There were notable variations in the timing of viability testing, with some trials conducting it before administration [28,47] and others just before cryopreservation [52]. As cryopreservation can be cytotoxic and only small adjustments in the vulnerable thawing process can have markedly negative MSC effects, the reported viability percentages may be questioned.

Of the trials that reported their post-cryopreservation thawing procedure, the following can be mentioned:

Kuah et al. (2018) thawed their AD-MSCs and drew them into a syringe and administered them for clinical therapy. Thus, the thawing solution and final vehicle medium consisted of cell culture media and cryopreservative. Cell viability prior to administration was not specified [27].

Yokota et al. (2019) rapidly thawed their AD-MSCs and then reconstituted them in Ringer’s solution immediately before direct intra-articular injection but did not report cell viability [45].

Freitag et al. (2019) reported AD-MSC viabilities after post-thaw handling of >90%. The cells underwent thawing through a sterile water bath, followed by removal of cryoprotectant via centrifugation, and washed in chilled phosphate-buffered saline (PBS). The cell pellet was finally resuspended in isotonic (0.9%) saline [28].

The most frequently utilized final vehicle medium in the clinical trials was either isotonic saline [28,29] alone or in combination with serum/plasma [38,47].

#### 1.3.3. Reconstitution to Low Cell Concentrations

Another conclusion from our previous work [84] was that the reconstitution of MSCs to low concentrations resulted in major cell loss and reduced viability. Using Ringer’s acetate as both a reconstitution solution and diluent, we examined the effect of final cell concentration post-reconstitution. Following thawing, MSCs were reconstituted and diluted to various concentrations, ranging from 5 × 10^6^ cells/mL to 1.3 × 10^5^ cells/mL. Total cell count and viability were assessed immediately after dilution and after 2 h of storage at RT. MSC suspensions diluted to concentrations below 8 × 10^5^ MSCs/mL exhibited significant cell loss, reaching 45%, along with a corresponding decrease in cell viability. This cell loss and poor viability was observed directly after dilution and further worsened after 2 h. However, cell loss and decreased viability were avoided by addition of serum or another protein-containing solution (i.e., culture medium or saline-HSA).

We are not aware of other publications that have considered the effect of reconstituting to low concentrations of MSCs.

Two of the included clinical trials employed low doses of MSCs. Pers et al. (2016), in their dose-finding study, treated 18 patients with moderate–severe knee OA with a low (0.4 × 10^6^ MSCs/mL), mid (2 × 10^6^ MSCs/mL) and high dosage (10 × 10^6^ MSCs/mL) of AD-MSCs. Viability testing just before clinical application showed ≥90% viability, even for the low-dose group. Of note, Pers et al. reconstituted the cells in a 3.6% human albumin and polyionic solution containing glucose, which, in line with our data, may have protected from cell loss and decreased viability [53].

Similarly, Chahal et al. (2019) divided 12 patients with moderate–severe knee OA into four different dosage groups and treated them with BM-MSCs. The groups were: low-dose (0.125 × 10^6^ MSCs/mL), mid-dose (1.25 × 10^6^ MSCs/mL), high-dose (6.25 × 10^6^ MSCs/mL), and a mixed-dose group receiving either 0.125, 1.25 or 6.25 × 10^6^ MSCs/mL. Viability analysis performed at harvest showed >70% but viability was not retested after reconstitution. The group formulated their cells in 6.5 + 1.5 mL (2.5% of the patient’s autologous serum in Plasma-Lyte A) before clinical application and thus may have avoided further decreasing cell viabilities [44].

*Summary of paragraph:* Several preclinical ex vivo studies have tested different pre-injection reconstitution solutions of freshly harvested cells. These studies found that MSCs can be stored for several hours in balanced isotonic salt solutions.

Clinical trials using freshly harvested MSCs for OA therapy have primarily utilized balanced isotonic salt solutions with or without serum/protein with high cellular viabilities >85%. When utilizing cryopreserved MSCs, preclinical ex vivo studies have identified that the presence of protein or serum in the thawing medium is crucial for ensuring high cellular viabilities. After thawing, MSCs can be washed free from unwanted solutions and reconstituted in the final vehicle medium. Both preclinical ex vivo studies and clinical trials have frequently utilized isotonic saline alone or in combination with serum/plasma as a final vehicle medium. Caution must be taken when using MSCs at low concentrations (<8 × 10^5^ MSCs/mL) as this can result in significant cell loss and poor viability, though it can be avoided by the addition of a serum/protein-containing solution.

### 1.4. Injection Procedure

#### Cannula Gauge Size

As a final possible viability influencing factor, MSCs are injected under pressure through a narrow cannula.

In preclinical ex vivo studies, the gauge size (G) was investigated in MSCs of human and animal origin.

BM-MSCs viability passed through 24 G, 25 G and 26 G cannulas were analyzed and compared to that of control cells only ejected through a syringe without any cannula. BM-MSC viability were high (>95%) after passing through the syringe in all groups. There was no significant difference between the three cannula groups and the control [85].

As larger caliber cannulas (19–21 G) are typically utilized for intra-articular or tendonous injections, viability analysis was also performed in this range.

Equine BM-MSCs were passed through 19 G, 20 G and 21 G cannulas and compared to control cells that did not pass a syringe. High and comparable BM-MSC viabilities were observed in all groups [86].

In our own unpublished work, we utilized AD-MSCs passed through a 21 G cannula directly after reconstitution (baseline) and again after 4 h of storage. Viability analysis was performed at baseline and after 4 h of storage and compared to unpassed AD-MSCs, and we found a viability range of 85–90% between all groups with no statistically significant differences.

Most clinical trials included failed to report the cannula gauge size used. Of those who reported size, this varied from 19 to 25 G, with the majority using 20 G, which, as presented, should also ensure high cellular viability.

*Summary of paragraph:* Preclinical studies have tested whether the cannula gauge size affects MSC viability. Cannulas from 19 G to 26 G were tested and none of the gauge sizes influenced MSC viability. Clinical trials have frequently used 20 G cannulas.
biomedicines-13-00509-t004_Table 4Table 4Quality control measures during MSC manufacturing.No.AuthorMSC Type Transplantation TypeMicrobiological Testing and Donor Transmission Status Confluency % Before HarvestPassage No.Morphology During GrowthSurface Marker Identity MSC ViabilityPotency Assay Genetic Stability 1Kim et al. [38]AD-MSCAutoSterility = Neg Mycoplasma = Neg Endotoxins = Neg90%3NSPositive (CD73, CD90)/negative (CD31, CD34, CD45)>80%NSNS2Günay et al. [39]UC-MSCAlloSterility = Neg Mycoplasma = Neg Endotoxins = Neg (<0.5 EU/mL) Donor transmission status stated as healthy. Unclear whether testing had been performed.80–90%2Cells were adherent to plastic.Positive (CD44, CD90, CD73, CD105)/negative (CD14, CD11b, CD45, HLA-DR)93.81% ± 0.98%NSTelomerase enzyme activity analysis3Samara et al. [40]UC-MSCAlloSterility = Neg Mycoplasma = Neg Endotoxins = Neg HIV = Neg HBV = Neg HCV = NegNS3Cells were adherent to plastic.Positive (CD44, CD90, CD73, CD105)/negative (CD14, CD19, CD31, CD34, CD45, HLA-DR)80%Multi-lineage differentiation Adipose = Pos Chondrogenic = Pos Osteogenic = PosKaryotype analysis4Kim et al. [30] Same as [38]





5Jianrui et al. [41]BM-MSCAutoNSNS2NSNSNSNSNS6Lu et al. [42]AD-MSCAlloSterility = Neg Mycoplasma = Neg Endotoxins = Neg (<0.4 EU/mL) HIV = Neg HBV = Neg HCV = Neg90%4Cells were adherent to plastic and were spindle-shaped with large oval nuclei.Positive (CD90, CD73, CD105)/negative (HLA-DR, CD14, CD45)>80%Multi-lineage differentiation Adipose = Pos Chondrogenic = Pos Osteogenic = PosNS7Dilogo et al. [43]UC-MSCAlloNS90%NSCells were adherent to plastic.NSNSNSNS8Chahal et al. [44]BM-MSCAutoSterility = Neg Mycoplasma = Neg Endotoxins = Neg80–90%4Cells were adherent to plastic and had a spindle-shaped morphology.Positive (CD90, CD73, CD105,)/negative (CD14, CD19, CD34, CD45, HLA-DR)>70%Multi-lineage differentiation Adipose = Pos Chondrogenic = Pos Osteogenic = Pos Licensed BM-MSC anti-inflammatory gene expression and TSG-6 protein profile.NS9Lu et al. [26]Same as [50]





10Lee et al. [29] Same as [38]





11Freitag et al. [28]AD-MSCAutoSterility = Neg Mycoplasma = Neg Endotoxins = Neg80%2NSPositive (CD90, CD73, CD105)/negative (CD14, CD19, CD34, CD45)>90%NSNS12Yokota et al. [45]AD-MSCAutoSterility = Neg Mycoplasma = Neg Endotoxins = Neg80%4Cells were adherent to plastic.NSNSNSNS13Zhao et al. [46]AD-MSCAlloSterility = Neg Mycoplasma = Neg Endotoxins = Neg (<0.4 EU/mL) HIV = Neg HBV = Neg HCV = Neg Syphilis = Neg85–90%4Adherent to plastic, spindle shape with large oval nuclei.Positive (CD90, CD73, CD105)/negative (HLA-DR, CD14, CD45)>80%Multi-lineage differentiation Adipose = Pos Chondrogenic = Pos Osteogenic = Pos Inhibition of T cell proliferation.Short tandem repeat analysis 14Soltani et al. [25]UC-MSCAlloSterility = Neg Mycoplasma = Neg HIV = Neg HBV = Neg HCV = Neg CMV= Neg80%12Adherent to plastic.Positive (CD90, CD73, CD105)/negative (CD45, CD31, CD34)NSMulti-lineage differentiation Adipose = Pos Osteogenic = PosKaryotype analysis15Bastos et al. [34] Same as [48]





16Matas et al. [47]UC-MSCAlloSterility = Neg Mycoplasma = Neg Endotoxins = Neg (<0.5 EU/mL) HIV = Neg HBV = Neg HCV = Neg Syphilis = Neg CMV= Neg70–80%5Plastic adherence.Positive (CD90, CD73, CD105)/negative (CD45, CD34, CD14, HLA-DR)>80%Multi-lineage differentiation Adipose = Pos Chondrogenic = Pos Osteogenic = Pos thrombospondin-2 (TSP2) enzyme-linked immunosorbent assay (chondrogenic differentiation).NS17Bastos et al. [48]BM-MSCAutoNS70–80%2Cells were adherent to plastic and displayed a typical myofibroblast pattern of growth. At the edge of the colonies, cells presented a stellate morphology, and at the core of the colonies, cells presented a fusiform monolayer distribution.Positive (CD90, CD73, CD146)/negative (CD14, CD31, CD34, CD45)NSNSNS18Emadedin et al. [49] Same as [62]





19Song et al. [50]AD-MSCAutoPreclinical toxicity in BALB/c-nu nude mice—no infections/reactions found. 90%4Cells were adherent to plastic and were spindle-shaped with a fibroblast-like morphology.Positive (CD90, CD73, CD29, CD49d)/negative (actin, HLA-DR, CD14, CD34, CD45)NSMulti-lineage differentiation Adipose = Pos Chondrogenic = Pos Osteogenic = PosPreclinical chronic tumorigenicity in BALB/c-nu nude mice—results indicated that AD-MSC did not induce tumor.20Spasovski et al. [51]AD-MSCAutoSterility = Neg Mycoplasma = NegNS2–3Cells were adherent to plastic and had a fibroblast-like morphology.Positive (CD90, CD73, CD105)/negative (CD34, CD45)>90%Multi-lineage differentiation Chondrogenic = Pos Osteogenic = PosNS21Kuah et al. [27]AD-MSCAlloNSNSNSNSNSNSNSNS22Al-Najar el al [24]BM-MSCAutoSterility = Neg Mycoplasma = Neg Endotoxins = Neg70–80% ≤4Cells were adherent to plastic.Positive (CD44, CD90, CD73, CD105)/negative (CD11b, CD19, CD34, CD45, HLA-DR)NSMulti-lineage differentiation Adipose = Pos Osteogenic = PosNS23Gupta et al. [52]BM-MSCAlloSterility = Neg Mycoplasma = Neg HIV = Neg HBV = Neg HCV = NegNSNSCells were adherent to plastic and had a fibroblastic and spindle-shaped morphologyPositive (CD90, CD73, CD105, CD 166)/negative (CD14, CD19, CD34, CD45, CD133, HLA-DR)≥85%Multi-lineage differentiation Adipose = Pos Chondrogenic = Pos Osteogenic = PosKaryotype analysis 24Pers et al. [53]AD-MSCAutoSterility = Neg Mycoplasma = Neg Endotoxins = NegNS1Cells were adherent to plastic.Positive (CD90, CD73, CD105)/negative (CD14, CD34, CD45)≥90%NSKaryotype analysis and measurement of human telomerase reverse transcriptase mRNA contents by qRT-PCR.25Soler et al. [54]BM-MSCAutoSterility = Neg Mycoplasma = Neg Endotoxins = Neg70–90%NSCells were adherent to plastic and had a fibroblastic morphology.Positive (CD90, CD73, CD105, HLA I)/negative (CD31, CD45, HLA II)85%NSNS26Lamo-Espinosa et al. [55,56]BM-MSCAutoNS70–80%NSCells were adherent to plastic.Positive (CD90, CD73, CD44)/negative (CD34, CD45)NSNSNS27Vega et al. [57]BM-MSCAlloHIV = Neg HBV = Neg HCV = Neg Syphilis = Neg80%2Cells were adherent to plastic and had a fibroblastic morphology.Positive (CD90, CD73, CD105, CD166)/negative (CD34, CD45, CD14, CD19, HLA II, HLA-DR)>98%NSNS28Jo et al. [58,59]AD-MSCAutoSterility = Neg Mycoplasma = Neg Endotoxins = Neg Preclinical toxicity in SCID mice—no infections/reactions found.90%3Cells were adherent to plastic and had a fibroblastic and spindle-shaped morphology.Positive (CD90, CD73)/negative (CD31, CD34, CD45)>95%Multi-lineage differentiation Adipose = Pos Chondrogenic = Pos Osteogenic = Pos Myoblasts = Pos Neuronal cells = PosKaryotype analysis and SNP genotyping. Preclinical tumorigenicity in mice—results indicated that AD-MSC did not induce tumor.29Orozco et al. [60,61]BM-MSCAutoNS80%3Cells were adherent to plastic and had a fibroblastic morphology.Positive (CD90, CD105, CD106, CD166, KDR)/negative (CD34, CD45, HLA-DR)91% ± 6%NSNS30Emadedin et al. [62]BM-MSCAutoSterility = Neg NS2Cells were adherent to plastic and had a fibroblast-like morphology.Positive (CD90, CD73, CD105, CD44) NSNSNS31Davatchi et al. [63,64]BM-MSCAutoSterility = Neg NSNSCells were adherent to plastic and had a fibroblast-like morphology.Positive (CD13, CD44, CD105)/negative (CD31, CD34, CD45)NSNSNS32Centeno et al. [65]BM-MSCAutoNS80%2–7Cells were adherent to plastic and had a spindle-shaped morphology.NSNSNSNS33Centeno et al. [66]BM-MSCAutoNSNS5Cells were adherent to plastic.NSNSNSNSAD-MSC = Adipose-derived mesenchymal stromal cell, Allo = Allogeneic, Auto = Autologous, BM-MSC = Bone marrow-derived mesenchymal stromal cell, CMV = Cytomegalovirus, EU/mL = Endotoxin unit/milliliters, HBV = Hepatitis B virus, HCV = Hepatitis C virus, HIV = Human immunodeficiency, mRNA = Messenger RNA, Neg = Negative, No. = Number, NS = Not specified, Pos = Positive, qRT-PCR = Quantitative reverse transcription polymerase chain reaction, SCID = Severe combined immunodeficient mice, SNP = Single-nucleotide polymorphism, Sterility = Testing for fungal and bacterial contamination (aerobic/anaerobic), UC-MSC = Umbilical cord-derived mesenchymal stromal cell.


### 1.5. Quality Control Measures

Requirements for quality control measures for drug products are detailed by the International Council for Harmonization of Technical Requirements for Pharmaceuticals for Human Use [70]. Additionally, guidelines for ATMP producers establish key standards for conducting stability studies, defining impurity-testing thresholds, and ensuring quality through GMP risk management.

Quality control measures are applied in the following steps: (1) starting material selection, (2) manufacturing process control, and (3) final product release [70].

The majority of recent clinical trials reported quality control measures at the three different levels. In contrast, earlier clinical trials often lacked such reporting, which may indicate a growing awareness and increased involvement of regulatory authorities in recent years.

#### 1.5.1. Donor Transmission Status

As for the starting material, especially when employing allogeneic transplantations, strict donor selection criteria must be ensured, and the donor transmission status must be verified. The clinical trials that used allogeneic transplantations tested their donors to be absent of pathogens. Most trials tested for human immunodeficiency (HIV), hepatitis B virus (HBV), and hepatitis C virus (HCV) [40,42,52], and a few extended the testing to include syphilis [46,57] and cytomegalovirus [25,47].

#### 1.5.2. MSC Viability

A critical step to live cellular therapies is to ensure the viability of the MSCs. Viability analysis should be performed in all steps of the manufacturing process and before administration. Viability testing was performed in the clinical trials included either by flow-cytometry or trypan blue cell staining and direct counting in Bürker-Türk counting chambers. Viability analysis was performed at different stages during MSC manufacturing and ranged between 70–98%.

#### 1.5.3. Passage and Genetic Stability

Passaging of the MSCs and the caution which must be exerted when passaging above passage 6 to 9 has also already been discussed above and resides in the possible cellular senescence which may be induced. The majority of the clinical trials passaged cells to P4–5. Regulatory agencies have previously accepted the safety of not exceeding higher passaging but are becoming extensively strict on demanding genetic stability analysis. Of the clinical trials included, a number performed karyotype analysis [25,40,52,53,58], telomerase enzyme activity analysis [39] or short tandem repeat analysis [46] to address this.

#### 1.5.4. Confluency

Preventing overconfluency ensures cell viability. The clinical trials included typically expanded the MSCs to 70–90% confluency before detaching the cells and splitting into further cultivation or preparing either for storage or fresh use. By detaching the cells from their plastic adherence at this percentage level before overgrowth, cell-to-cell contact inhibition is avoided [70].

#### 1.5.5. Morphology During Growth

The International Society for Cellular Therapy (ISCT) recommends assessing morphology by microscopy and evaluating surface marker identity and multi-lineage differentiation potential [87].

The majority of the clinical trials included adhere to the ISCT recommendation on reporting the MSC ability to adhere to plastic during cultivation. Some trials additionally describe the appearance as fibroblastic and with a spindle-shaped morphology [50,52,58].

#### 1.5.6. Surface Marker Identity

The ISCT recommends flow-cytometric characterization of the cellular product to ensure MSC authenticity. MSCs must express CD105, CD73 and CD90, and lack expression of CD45, CD34, CD14 or CD11b, CD79α or CD19 and HLA-DR surface molecules [87]. The majority of the included clinical trials assessed the MSC identity for the positive markers. A lack of testing was observed towards the negative markers, especially CD11b and CD79α.

#### 1.5.7. Potency Assay

A potency assay measures the MSCs biological activity. It directly relates to efficacy, correlating with the intended use and predicting the therapeutic effect.

As part of the ISCT recommendation, the MSCs multi-lineage differentiation abilities must also be assessed [87]. MSCs are able to differentiate into osteoblasts, adipocytes and chondroblasts in vitro and, as the chondrogenic potential is desired in the osteoarthritic knee, the differentiation potential can serve as a potential potency assay. A total of 11 out of the 33 included clinical trials performed multi-lineage differentiation assays and all found their MSCs to be able to differentiate into osteoblasts, adipocytes and chondroblasts. Three of the included clinical trials also performed further potency testing. One employed enzyme-linked immunosorbent assays to assess the chondrogenic potential [47]. For the anti-inflammatory abilities, inhibition of T cell proliferation [46] and anti-inflammatory gene expression [44] analysis was carried out. A standardized potency assay to demonstrate the immunomodulatory or regenerative capabilities of the MSC has not yet been developed but extensive work is being undertaken in this field.

#### 1.5.8. Microbiological Testing

Sterility testing should be performed as late in the manufacturing process as possible but may be relevant to include in the process as well. Testing must include aerobic/anaerobic bacteria and fungi together with mycoplasma and endotoxins. The majority of the included trials performed sterility testing (for fungal and bacterial (aerobic/anaerobic) contamination) and, in trials after 2013, testing for mycoplasma and endotoxin was also carried out.

*Summary of Paragraph*: Quality control measures are implemented in all levels of MSC manufacturing, from the starting material to production and final release before clinical administration.

When utilizing allogeneic transplantations, donors are screened for infectious diseases such as HIV, HBV, HCV, syphilis and cytomegalovirus.

The final drug product should always be tested for microbial contamination. Additionally, microbial contamination testing should also be conducted on relevant intermediates throughout the manufacturing process.

Sterility testing includes fungal and bacterial (aerobic/anaerobic) contamination and testing for mycoplasma and endotoxin.

A critical step in live cellular therapies is to ensure the viability of the MSCs, and this ought to be performed at several stages during the manufacturing process, and, in particular, prior to administration. MSCs are usually passaged to passage 4–5, which is below the limit of concern for possible cellular senescence. Regulatory authorities demand stricter testing, including genetic stability testing, which may include karyotype analysis.

MSC morphology should be microscopically observed as being adherent to plastic and having a fibroblastic and spindle-shaped morphology.

The cellular flow-cytometric surface marker identity of the MSCs is also analyzed. MSCs typically express CD105, CD73 and CD90, and lack expression of CD45, CD34, CD14, CD19 and HLA-DR surface molecules.

Finally, the biological activity of the MSCs may be assessed by employing a potency assay. The MSCs multi-lineage differentiation into osteoblasts, adipocytes and chondroblasts in vitro has frequently been analyzed; however, a more specific potency assay has not yet been developed.

### 1.6. Minimal Reporting Criteria for Future Clinical Trials

As seen in Table 2, Table 3 and Table 4, there is a noticeable lack of standardization in reporting key aspects of MSC manufacturing. Specifically, this includes details regarding MSC cultivation, pre-injection reconstitution, and the clinical formulation applied in MSC-based therapies for knee OA. To address this gap, we propose a set of minimal reporting criteria to guide future clinical trials (Table 5).
biomedicines-13-00509-t005_Table 5Table 5Minimal reporting criteria for future clinical trials.CategoryInformation to Report**Cellular Product Characteristics**
MSC SourceSpecify tissue origin (e.g., adipose, bone marrow, umbilical)Transplantation SettingAutologous or allogeneicCell StateFreshly harvested, cryopreserved, or combination**Culture Process**
Tissue ProcurementMethod of source tissue acquisitionCell IsolationTechnique for extracting MSCs from source tissueExpansion MethodProtocol for MSC culturingCulture MediumExact composition of expansion mediumGrowth SupplementsAdditives in expansion mediumGrowth ConditionsEnvironmental parameters for cell expansionHarvesting MethodTechnique for collecting expanded MSCsConfluence at HarvestDegree of cell density upon collectionPassage NumberFinal passage at harvestingStorage MethodCryopreservation protocol or fresh transport protocolViability AssessmentCell viability percentage and testing stage**Pre-injection Reconstitution**
Thawing ProtocolThawing medium and possible serum addition reporting (autologous or allogeneic) and procedure for thawing cryopreserved cellsFinal Vehicle MediumComposition of carrier solution for MSCs and reporting of serum addition (autologous or allogeneic)Vehicle Medium VolumeQuantity of carrier solution usedTotal MSC DoseNumber of cells used for therapyMSC ConcentrationCells per mL in final preparation**Injection Procedure**
Cannula GaugeSize of needle used for intra-articular injectionInjection TechniqueDetailed procedure for MSC delivery to the knee**Quality control measures**
Microbiological testing Contamination testing and testing stageConfluence at HarvestDegree of cell density upon collectionPassage Number Final passage at harvestingMorphological AssessmentDescription of MSC morphology during growthSurface Marker ProfileCellular identity markersViability AssessmentCell viability percentage and testing stagePotency AssayFunctional assay linked to the efficacyGenetic stability Genetic analysesMSC = Mesenchymal stromal cell.


## 2. Conclusions

This review examines the manufacturing methods utilized in producing ex vivo expanded MSC products for knee OA therapy. It highlights the significant challenges in standardizing MSC production for clinical applications despite their increasing use. These challenges arise from variability in tissue sources, procurement techniques, and manufacturing processes, further exacerbated by insufficient reporting of production protocols.

To address these issues, this review evaluates the entire production pipeline—from tissue procurement and cell culture to reconstitution—with a focus on optimizing processes to ensure high MSC yield, viability, and stability. Additionally, it proposes a set of minimal reporting criteria to guide future clinical trials.

By adopting these recommendations, the standardization and reproducibility of MSC production across laboratories and clinical trials can be greatly enhanced, advancing the field of regenerative medicine for knee OA as well as other disease settings.

## Figures and Tables

**Figure 1 biomedicines-13-00509-f001:**
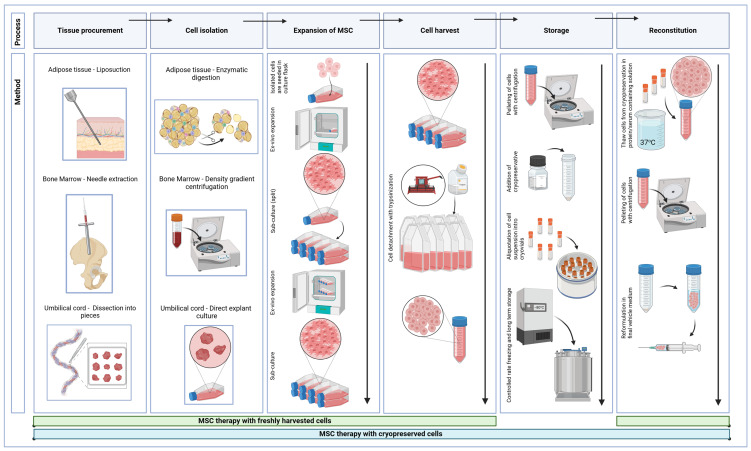
Schematic overview of the MSC manufacturing process. Step-by-step production of MSCs from tissue procurement and isolation from various tissue sources to expansion and cell harvesting. Production is illustrated for both fresh and cryopreserved MSCs (green bar showing the process for MSC therapy with fresh cells, blue bar showing it for cryopreserved cells). (MSC = Mesenchymal stromal cell). Biorender was used to create Figure 1.

**Table 1 biomedicines-13-00509-t001:** Overview of clinical trials included.

No.	Author	Country	Year	Study Design	Sample Size	Treatment/Control (n)	Intervention	Total MSC	Control	Follow-Up (Months)	MSC Type	Transplantation Type
1	Kim et al. [38]	South Korea	2023	RCT	261	131/130	sIAI of AD-MSC	100 × 10^6^	sIAI of 2.1 mL of normal saline and 0.9 mL auto serum	6	AD-MSC	Auto
2	Günay et al. [39]	Turkey	2022	CS	10	10	sIAI of UC-MSC	100 × 10^6^	NA	12	UC-MSC	Allo
3	Samara et al. [40]	Jordan	2022	Prospective open-label clinical trial	16	16	2 × IAI of UC-MSC, interval of 1 month	First injection: 34–50 × 10^6^, Second injection: 33–55 × 10^6^	NA	48	UC-MSC	Allo
4	Kim et al. [30]	South Korea	2022	CS	11	11	sIAI of AD-MSC	100 × 10^6^	NA	60	AD-MSC	Auto
5	Jianrui et al. [41]	China	2020	Retrospective study	86	40/46	3 × IAI of BM-MSC, interval of 1 month	NS	AD followed by 5 × IAI of 2 mL HA, interval of one week between injections	12	BM-MSC	Auto
6	Lu et al. [42]	China	2020	Double-blinded, dose-ranging RCT	22	(1) low-dose group n = 7, (2) mid-dose group n = 8, (3) high-dose group n = 7	2 × IAI of AD-MSC, for all dose groups, interval of 3 weeks	(1) low-dose group (10 × 10^6^), (2) mid-dose group (20 × 10^6^), (3) high-dose group (50 × 10^6^)	NA	12	AD-MSC	Allo
7	Dilogo et al. [43]	Indonesia	2020	Prospective open-label clinical trial	29	29	3 × IAI, 1 injection UC-MSC, 2nd and 3rd injection 2 mL HA, interval of 1 week	10 × 10^6^	NA	12	UC-MSC	Allo
8	Chahal et al. [44]	Canada	2019	Prospective, open-label, dose-escalating clinical trial	12	(1) low-dose group n = 3, (2) mid-dose group n = 3, (3) high-dose group n = 9, mixed-dose group (one patient low dose, one mid dose and one high dose)	sIAI of BM-MSC for all groups	(1) low-dose group (1 × 10^6^), (2) mid-dose group (10 × 10^6^), (3) high-dose group (50 × 10^6^), (4) mixed-dose group (1, 10, 50 × 10^6^)	NA	12	BM-MSC	Auto
9	Lu et al. [26]	China	2019	RCT	53	26/27	2 × IAI of AD-MSC and 2 sham injections	50 × 10^6^	4 × IAI of 2.5 mL HA	12	AD-MSC	Auto
10	Lee et al. [29]	South Korea	2019	RCT	24	12/12	sIAI of AD-MSC	100 × 10^6^	sIAI of 3 mL saline	6	AD-MSC	Auto
11	Freitag et al. [28]	Australia	2019	RCT	30	(1) group n = 10, (2) group n = 10/10	(1) group sIAI of AD-MSC, (2) group 2 × IAI of AD-MSC, interval of 6 months	100 × 10^6^	Conventional management only	12	AD-MSC	Auto
12	Yokota et al. [45]	Japan	2019	CS	80	(1) group n = 42, (2) group n = 38	(1) group sIAI of AD-MSC or (2) group sIAI of 5 mL SVF	12.8 × 10^6^ ± 2.9 × 10^6^	NA	6	AD-MSC	Auto
13	Zhao et al. [46]	China	2019	Double-blinded RCT	18	(1) low-dose group n = 6, (2) mid-dose group n = 6, (3) high-dose group n = 6	2 × IAI of AD-MSC for all dose groups, interval of 3 weeks	(1) low-dose group (10 × 10^6^ cells), (2) mid-dose (20 × 10^6^) (3) high-dose group (50 × 10^6^)	NA	12	AD-MSC	Allo
14	Soltani et al. [25]	Iran	2019	RCT	20	10/10	sIAI of UC-MSC	50–60 × 10^6^	sIAI of 10 mL saline	6	UC-MSC	Allo
15	Bastos et al. [34]	Brazil	2019	RCT	47	(1) group n = 16, (2) group n = 14/17	(1) group sIAI of BM-MSC, (2) group sIAI of BM-MSC + 10 mL auto PRP	40 × 10^6^	sIAI of corticosteroid (4 mg of de × amethasone)	12	BM-MSC	Auto
16	Matas et al. [47]	Chile	2018	RCT	29	(1) group n = 10, (2) group n = 10/9	(1) group 2 × IAI of UC-MSC (interval of 6 months), (2) group 2 × IAI of UC-MSC (baseline) and 5% AB plasma in 3 mL saline (6 months)	20 × 10^6^	2 × IAI of 3 mL HA (interval 6 months)	12	UC-MSC	Allo
17	Bastos et al. [48]	Brazil	2018	RT	18	(1) group n = 9, (2) group n = 9	(1) group sIAI of BM-MSC, (2) group sIAI of BM-MSC+ 10 mL auto PRP	40 × 10^6^	NA	12	BM-MSC	Auto
18	Emadedin et al. [49]	Iran	2018	RCT	47	22/25	sIAI of BM-MSC	40 × 10^6^	sIAI of 5 mL saline supplemented with 2% HSA	6	BM-MSC	Auto
19	Song et al. [50]	China	2018	Double-blinded, dose-ranging RCT	18	(1) low-dose group n = 6, (2) mid-dose group n = 6, (3) high-dose group n = 6	3 × IAI for all dose groups, interval of 3 weeks and the last injection after 6 months	(1) low-dose group (10 × 10^6^), (2) mid-dose group (20 × 10^6^), (3) high-dose group (50 × 10^6^)	NA	24	AD-MSC	Auto
20	Spasovski et al. [51]	Serbia	2018	CS	9	9	sIAI of AD-MSC	5–10 × 10^6^	NA	18	AD-MSC	Auto
21	Kuah et al. [27]	Australia	2018	RCT	20	(1) low-dose group n = 8, (2) high-dose group n = 8/4	sIAI of AD-MSC for all dose groups	(1) low-dose group (3.9 × 10^6^), (2) high-dose group (6.7 × 10^6^)	sIAI of 2 mL cell culture media and cryopreservative	12	AD-MSC	Allo
22	Al-Najar et al. [24]	Jordan	2017	Prospective open-label clinical trial	13	13	2 × IAI of BM-MSC, interval of 4 weeks	30.5 × 10^6^	NA	24	BM-MSC	Auto
23	Gupta et al. [52]	India	2016	RCT	60	(1) dose level I group n = 10, (2) dose level II group n = 10, (3) dose level III group n = 10, (4) dose level IV group n = 10/20	sIAI of BM-MSC for all dose groups followed by sIAI injection of 2 mLHA (20 mg)	(1) dose level I group (25 × 10^6^), (2) dose level II group (50 × 10^6^), (3) dose level III group (75 × 10^6^), (4) dose level IV group (150 × 10^6^)	(5) 10 patients sIAI of 2 mLPLASMA-LYTE A, (6) 10 patients sIAI of 4 mLPLASMA-LYTE A, for all 20 patients this was followed by sIAI injection of 2 mLHA (20 mg).	12	BM-MSC	Allo
24	Pers et al. [53]	France and Germany	2016	Prospective, open-label, dose-escalating clinical trial	18	(1) low-dose group n = 6, (2) mid-dose group n = 6, (3) high-dose group n = 6	sIAI of AD-MSC for all dose groups	(1) low-dose group (2.0 × 10^6^), (2) mid-dose group (10 × 10^6^), (3) high-dose group (50 × 10^6^)	NA	6	AD-MSC	Auto
25	Soler et al. [54]	Spain	2016	Prospective open-label clinical trial	15	15	sIAI of BM-MSC	40.9 × 10^6^	NA	12 (48)	BM-MSC	Auto
26	Lamo-Espinosa et al. [55,56]	Spain	2016/2018	RCT	30	(1) low-dose group n = 10, (2) high-dose group n = 10/10	sIAI of BM-MSC for both dose groups followed by sIAI injection of 4 mLHA (60 mg)	(1) low-dose group (10 × 10^6^), (2) high-dose group (100 × 10^6^)	sIAI of 4 mL HA (60 mg)	12/48	BM-MSC	Auto
27	Vega et al. [57]	Spain	2015	RCT	30	15/15	sIAI of BM-MSC	40 × 10^6^	sIAI of 3 mL HA (60 mg)	12	BM-MSC	Allo
28	Jo et al. [58,59]	South Korea	2014/2017	CS	18	(1) low-dose group n = 3, (2) mid-dose group n = 3, (3) high-dose group n = 12	sIAI of AD-MSC	(1) low-dose group (10 × 10^6^), (2) mid-dose group (50 × 10^6^), (3) high-dose group (100 × 10^6^)	NA	6/48	AD-MSC	Auto
29	Orozco et al. [60,61]	Spain	2013/2014	CS	12	12	sIAI of BM-MSC	40 × 10^6^	NA	12/24	BM-MSC	Auto
30	Emadedin et al. [62]	Iran	2012	Case study	6	6	sIAI of BM-MSC	20–24 × 10^6^	NA	12	BM-MSC	Auto
31	Davatchi et al. [63,64]	Iran	2011/2016	Case study	4/3	4/3	sIAI of BM-MSC	8–9 × 10^6^	NA	6/48	BM-MSC	Auto
32	Centeno et al. [65]	USA	2011	Case series of injection in patients with various orthopedic diseases with complaint of pain in the knee, hip, foot-ankle, shoulder, lower back, hand/wrist and various other musculoskeletal sites	Cohort 1 50/Cohort 2 290 (133 patients receiving knee injection as part of both cohort 1 and 2)	Cohort 1 n = 50/Cohort 2 n = 290	Some patients received multiple IAI of BM-MSC	Fresh: NS/Cryo: 1–3 × 10^6^	NA	36	BM-MSC	Auto
33	Centeno et al. [66]	USA	2008	Case report	1	1	IAI of BM-MSCs followed by 1 mL of auto nucleated cells suspended in PBS and 1 mL of 10% auto PL. The patient returned for 2 additional 10% intraarticular knee PL injections (1 mL) at week 1 and week 2 (post-transplantation). With the 2-week post-transplant PL supplementation, 1 mL of 10 ng/mL dexamethasone was also injected	22.4 × 10^6^	NA	6	BM-MSC	Auto

AD = Arthroscopic debridement, AD-MSC = Adipose-derived mesenchymal stromal cell, Allo = Allogeneic, Auto = Autologous, BM-MSC = Bone marrow-derived mesenchymal stromal cell, CS = Cohort study, HA = Hyaluronic acid, HSA = Human serum albumin, MSC = Mesenchymal stromal cell, NA = Not applicable, NS = Not specified, PBS = Phosphate-buffered saline, PL = Platelet lysate, PRP = Platelet-rich plasma, RCT = Randomized controlled trial, RT = Randomized trial, sIAI = Single intra-articular injection, SVF = Stromal vascular fraction, UC-MSC = Umbilical cord-derived mesenchymal stromal cell, USA = United States of America.

**Table 2 biomedicines-13-00509-t002:** Manufacturing process of MSCs in the clinical trials.

						MSC Production						
No.	Author	MSC Type	Fresh/ Cryo	Tissue Procurement	Cell Isolation	Expansion of MSC	Cell Harvest	Storage	Culture Medium	Confluency % Before Harvest	Pass-Age No.	MSC via-Bility
1	Kim et al. [38]	AD-MSC	Cryo	≈20 mL of adipose tissue was obtained by lipoaspiration from abdominal subcutaneous fat.	Aspirated tissues were digested with collagenase I (1 mg/mL) under gentle agitation for 60 min at 37 °C. The digested tissues were filtered through a 100-μm nylon sieve to remove cellular debris and were centrifuged to obtain a pellet. The pellet was resuspended in DMEM-based media containing 0.2 mM ascorbic acid and 10% FBS. The cell suspension was recentrifuged, and the pellet was collected.	The cell fraction was cultured for 4–5 days until confluent (passage 0). Then, the cells were split and further passaged (until P3).	NS	Cells were harvested at P3 and stored in liquid nitrogen (−196 °C).	Keratinocyte-SFM-based media 0.2 mM ascorbic acid, 0.09 mM calcium, 5 ng/mL rEGF, and 5% FBS	90%	3	>80%
2	Günay et al. [39]	UC-MSC	Cryo	The umbilical cord tissue from a healthy donor was collected.	Ten cm umbilical cord tissue was longitudinally cut, followed by washing with PBS. Wharton’s jelly tissue was isolated and diced into cubes for explant culture.	The sections of Wharton’s jelly tissue were placed into culture flasks and incubated in a humidified 37 °C incubator with 5% CO_2_.	At the end of primary culture, when the cells reached confluence, the cells were washed with PBS, detached with EDTA and 1 to 1.4 × 10^2^ cells/per flask were passaged.	Cells were cultured to P2 and cryopreserved. 5 days before the intra-articular transplantation, cells were thawed and re-plated. Cells were washed with saline and passed through 100 μm strainers before suspending the cells and administering them.	Alpha-MEM with 1% pen/strep and 10% allogeneic human serum	80–90%	2	93.81% ± 0.98%
3	Samara et al. [40]	UC-MSC	Cryo	Wharton jelly was donated from a full-term infant delivered by cesarean section.	Wharton’s jelly were disinfected and cut into 0.5–1 mm^2^ pieces. These pieces were transferred to 150 cm^2^ plates containing culture medium.	Cells were cultured in a humidified 37 °C incubator with 5% CO_2_. The medium was replaced after 6 days to allow cells to migrate from the explants.	Adherent confluent cells were harvested with TrypLE 10X and propagated at a seeding density of 4000 cells/cm^2^	Cells were cultured to P3 and cryopreserved in freezing bags with synth-afreeze.	Alpha-MEM supp. with 5% PL, 1% pen/strep 3U heparin, and 4 mML-glutamine	NS	3	80%
4	Kim et al. [30]			Same as [38]								
5	Jianrui et al. [41]	BM-MSC	Cryo	50 mL of bone marrow was extracted from the posterior superior iliac spine.	Nucleated cells were isolated by density gradient centrifugation.	Isolated cells were cultured and passaged.	Three days before the knee injection cells were thawed and cultured. On the day of injection, the cells were collected using 0.25% trypsin and centrifuged at 2000 rpm for 6 min. The supernatant was discarded and the cells were formulated in the final vehicle medium.	Cells at P2 were frozen and stored at −80 °C.	NS	NS	2	NS
6	Lu et al. [42]			Same as [50]								
7	Dilogo et al. [43]	UC-MSC	Fresh	Umbilical cord from at-term healthy delivery was collected.	Ten cm of umbilical cord was dissected and washed briefly in 0.5% povidone iodine (betadine) containing PBS, followed by washing in PBS to remove blood and betadine. Further, the umbilical arteries and vein were dissected and discarded, and the umbilical cord was minced in culture medium.	Wharton’s jelly explants were placed in a 24 well plate one piece per well and immersed in a small amount of culture medium and incubated in a humidified 37 °C incubator with 5% CO_2_. When the explant attached to the plastic, 200–500 µLof medium was added. Medium change was performed every 2–3 days, while explant was still attached. The successive cultures were observed daily to detect cell growth, and when the cells attained confluence, they were harvested. Therefore, one explant could be harvested several times.	When cells became confluent, they were harvested using TrypLE.	NA	Alpha-MEM and DMEM, pen/strep, amphoteri-cin B, 1% L-Glutamine, and 10% PL	90%	NS	NS
8	Chahal et al. [44]	BM-MSC	Fresh	50 mLof bone marrow was extracted from the posterior superior iliac spine.	Nucleated cells were isolated by density gradient centrifugation.	≈30 × 10^6^ of the isolated cells were plated in culture flasks containing culture medium and incubated (P0). After cells harvest (day 0), every 3–4 days the medium was changed and/or cells were passaged up to P3 or P4.	Cells were harvested by cell dissociation with TrypLE Select, washed three times (two times in Plasma-Lyte A and one time in excipient) and resuspended in the final vehicle medium.	NA	DMEM low glucose, 1% Glutamax, and 10% FBS	80–90%	3–4	>70%
9	Lu et al. [26]			Same as [50]								
10	Lee et al. [29]			Same as [38]								
11	Freitag et al. [28]	AD-MSC	Cryo	Up to 60 mL of adipose tissue was obtained by lipoaspiration from abdominal subcutaneous fat.	Lipoaspirate was processed to isolate the SVF through enzymatic digestion followed by centrifugation.	Cell culturing was performed within standard culture growth media (not specified) and during hypoxic conditions. Cells were passaged until P2.	NS	At P2, cells were washed three times to remove FBS and then cryopreserved.	Standard growth media containing 10% FBS	80%	2	>90%
12	Yokota et al. [45]	AD-MSC	Cryo	≈20 mL of adipose tissue was obtained by lipoaspiration from the lower abdominal subcutaneous fat.	Harvested adipose tissue was centrifuged at 800× *g* for 5 min. The adipose portion was isolated, lavaged with lactated Ringer’s solution and digested with collagenase and therymolysin at 37 °C for 45 min to liberate individual cells. Enzymatically digested tissue was centrifuged at 600× *g* for 5 min and the SVF was seeded in culture flasks.	Cells were cultured in culture medium in a humidified 37 °C incubator with 5% CO_2_. The medium was replaced the day after seeding and every 3 days thereafter.	When cells reached confluency, they were detached with Trypsin-EDTA and reseeded in culture flasks. Cells were passaged up to P4 over the course of 4 weeks for large-scale expansion.	Subconfluent cells in a proliferative state were collected in DMSO-free STEM-CELLBANKER^®^, aliquoted into sterile cryopreservation vials, visually inspected for particulate matter contamination, and frozen overnight at −80 °C. The frozen cells were then moved to a liquid nitrogen freezer (−150 °C) for long-term storage.	DMEM/Ham’s F-12 supp. with NeoSERA^®^ bovine PRPderived serum	80%	4	NS
13	Zhao et al. [46]	AD-MSC	Cryo	50 g of adipose tissue was obtained by lipoaspiration from the abdominal subcutaneous fat.	The lipoaspirate was washed three times with PBS to remove blood cells and tissue debris. Adipose tissues were digested with DMEM containing 0.1% *w*/*v* collagenase A type I under gentle agitation for 60 min at 37 °C. The digested tissues were centrifuged at 1000 rpm for 8 min to obtain a pellet and were filtered through a 100-μm nylon filter to remove cellular debris.	The cells were seeded in culture flasks and incubated in a humidified 37 °C incubator with 5% CO_2_. The cells were maintained in culture medium for 5–7 days until confluent (passage 0). After digestion with trypsin (0.125%) combined with EDTA (0.01%) for 1.5–2.5 min, cells were passaged at 5 × 10^3^/cm^2^ up to P4.	When the cells reached confluency, they were digested with trypsin (0.125%) combined with EDTA (0.01%) for 1.5–2.5 min and washed with PBS.	The cells were finally suspended in a serum-free cryopreservation solution at a cell concentration of 0.5–2 × 10^7^ cells/mL and immediately cryopreserved in liquid nitrogen.	Serum-free proliferation media	85–90%	4	>80%
14	Soltani et al. [25]	UC-MSC	Fresh	Full-term placentas were donated from healthy mothers who had normal vaginal delivery without complication.	Placenta (3–4 g) was rinsed and minced. The minced tissue was washed three times with saline to remove remaining blood, before being incubated with 1 mg/mL collagenase at 37 °C for 3 h, with shaking every 30 min. Then, saline was added, and the mixture was shaken and centrifuged.	The supernatant was discarded, and the cell pellet was cultivated in culture medium and placed in a humidified 37 °C incubator with 5% CO_2_. Primary cultures were maintained for 1 week in small, digested residues; non-adherent cells were removed by changing the culture medium. New medium was added twice weekly.	Cells were passaged, and when a sufficient number of cells was reached for further clinical applications TrypLE Express was used for harvest.	NA	DMEM supp. with 10% FBS	80%	12	NS
15	Bastos et al. [34]			Same as [48]								
16	Matas et al. [47]	UC-MSC	Cryo	Cords were obtained from healthy full-term donors that had given birth by cesarean section.	Umbilical cords were aseptically stored in sterile PBS supp. with 100 U/mL pen/strep. Within 3 h of birth, the umbilical cord was sectioned and washed with PBS and antibiotics. Cells were obtained from the Wharton’s jelly by dissection into small fragments (1–2 mm).	Small fragments containing cells were seeded in 100 mm culture plates and maintained in culture medium. At 48 h, non-adherent cells were removed and washed with PBS, and culture medium was replaced with fresh medium every 3 days.	When the cell culture reached confluency, cells were trypsinized with TrypLE TM Express and reseeded at a density of 2500 cells per cm^2^ into 500-cm^2^ flasks.	At P3, cells were cryopreserved in Profreeze. Before administration, cells were thawed and passaged until P5, then harvest and washed twice with PBS and suspended in final vehicle medium.	MEM and Alpha-MEM high glucose supp. with 10% heat-inactivated FBS, 1% pen/strep, and 2 mM L-glutamine	70–80%	5	>80%
17	Bastos et al. [48]	BM-MSC	Fresh	≈80–100 mL of bone marrow was extracted from the posterior superior iliac spine.	MNCs from bone marrow samples were separated using the Sepax automated closed system.	After isolation, cells were seeded at a density of 4 × 10^5^ cells/cm^2^ in culture medium and incubated for 5 days in a humidified 37 °C incubator with 5% CO_2_. Non-adherent cells were then discarded after medium exchange. Medium exchange was performed every 3–4 days.	Once the cells reached confluency, they were detached from culture flasks using 0.05% trypsin solution and re-seeded onto new culture flasks. Cultures were maintained for no more than P2.	NA	Alpha-MEM supp. with 10% FBS	70–80%	2	NS
18	Emadedin et al. [49]			Same as [62]								
19	Song et al. [50]	AD-MSC	Fresh	50 mL of adipose tissue was obtained by lipoaspiration from the abdominal subcutaneous fat.	Adipose tissues were digested with collagenase I (1 mg/mL) under gentle agitation for 60 min at 37 °C. Digested tissues were filtered through a 100 um nylon sieve to remove cellular debris and were centrifuged at 470× *g* for 5 min to obtain a pellet, which was resuspended in DMEM–based media containing 0.2 mM ascorbic acid and 10% FBS. The cell suspension was re-centrifuged at 470× *g* for 5 min. The supernatant was discarded, and the cell pellet was collected.	The cell fraction was cultured overnight in serum-free medium in a humidified 37 °C incubator with 5% CO_2_. After 24 h, non-adherent cells were removed by washing with PBS. Next, cells were maintained for 4–5 days until confluent (P0). When the cells reached confluency, they were passaged in serum-free medium.	Cells at P4 were collected and formulated in the final vehicle medium.	NA	Serum-free medium	90%	4	NS
20	Spasovski et al. [51]	AD-MSC	Fresh	5 mL of adipose tissue was obtained by excision from the abdominal subcutaneous fat.	The adipose tissue was left overnight at room temperature and processed the following day. After repeated washing in PBS, the tissue was treated with 0.1% collagenase until it was completely dissolved. Collagenase solution was neutralized by DMEM with low glucose supp. with 10% auto serum and antibiotic/antimycotic solution. Cells were filtered through a 100-μm filter.	Cells were seeded at 6 × 10^4^/cm^2^ in culture medium. After 1-week, non-adherent cells were washed away and cells were cultured for 2–3 weeks, until P2–3.	After trypsinization (trypsin-EDTA), cells were resuspended in the final vehicle medium.	NA	DMEM supp. with 10% auto serum and pen/strep	NS	2–3	>90%
21	Kuah et al. [27]	AD-MSC	Cryo	Cells used for this study were derived from a single human donor.	NA	Not noted beyond that cells were isolated, and culture expanded in a good manufacturing practice accredited facility.	NA	Cells were stored in a CryoVial^®^ and maintained at or below −150 °C prior to administration.	NS	NS	NS	NS
22	Al-Najar el al [24]	BM-MSC	Fresh	35–40 mL of bone marrow was extracted from the posterior superior iliac spine.	The bone marrow aspirates were diluted in a 1:1 ratio with PBS. MNCs were separated by density gradient centrifugation.	Separated cells were seeded at a density of 0.16 × 10^6^ cells/cm^2^ in tissue culture flask in culture medium. Cells were allowed to attach for 24 h before changing media. Subsequently, the culture medium was changed twice a week.	When cultures reached confluency, subculturing was performed using trypsin-EDTA 0.25%. After the primary passage, cells were seeded at a density of 4 × 10^3^ cells/cm^2^. Cells were cultured until reaching an average number of 30.5 × 10^6^ cells per dose injected per patient (*p* ≤ 4). For injection, MSCs were washed and suspended in the final vehicle medium.	NA	Alpha-MEM supp. with 100 IU pen/strep, 10% FBS and 2 mM L-glutamine	70–80%	≤4	NS
23	Gupta et al. [52]	BM-MSC	Cryo	The MSC product used consisted of pooled MSCs from three different healthy volunteers.	Bone marrow aspirate was diluted (1:1) with KO-DMEM and centrifuged at 1800 rpm for 10 min to remove the anti-coagulant. The supernatant was discarded, and the bone marrow was again diluted with KO-DMEM. MNCs were separated by the Ficoll density gradient method (1.077 g/mL density). MNCs accumulated on the Ficoll–plasma interface were isolated and washed again with KO-DMEM.	Isolated cells were plated into culture flasks and cultured in culture medium and placed into a humidified 37 °C incubator with 5% CO_2_. Non-adherent cells were removed after 48 h by replacing with fresh media. Subsequently the medium was replenished every 48 h.	Upon confluency, the cells were harvested with 0.25% trypsin–EDTA and replated at a density of 1000 cells/cm^2^. MSCs were harvested using 0.25% trypsin–EDTA and centrifuged at 1800 rpm for 10 min.	Cells were resuspended in a freezing solution containing 90% (*v*/*v*) sterile FBS and 10% DMSO. Cells were loaded in 2 mL cryovials at a concentration of 3 × 10^6^ cells/vial and frozen using a programmable slow freezing unit. After freezing, the cryovials were transferred in a liquid nitrogen vapor-phased cryo-container for long-term storage.	KO-DMEM supp. with 10% FBS, 2 mM glutamax, pen/strep	NS	NS	≥85%
24	Pers et al. [53]	AD-MSC	Fresh	>60 g of adipose tissue was obtained by lipoaspiration from the abdominal subcutaneous fat.	Aliquots of 10 g of adipose tissue were mixed with 34 mL of the collagenase solution and incubated at 37 °C for 45 min. Enzymatic digestion was stopped by the addition of culture medium. After homogenization, the digested suspension was passed through sterile 100-um filters. The cells were centrifuged at room temperature for 10 min at 600× *g*. The supernatant was discarded and the SVF was resuspended in 20 mL culture medium.	The cells from the SVF were then seeded in a culture chamber at a density of 4.103 cells/cm^2^ in culture medium. Cells were incubated in a humidified 37 °C incubator with 5% CO_2_. After an initial 24 h incubation, the non-adherent cells were removed. The adherent cells were washed once with PBS, and culture medium was added for 7 days. The medium was replaced at day 4 and day 6 of culture.	At day 8 (primary culture, P0), the cells were harvested with the following protocol: after aspiration of the medium and washing with PBS, 50 mL of irradiated trypsin solution was added for 5 min at room temperature. After trypsinization culture medium was added and the cells were collected in a transfer bag. The cells were seeded in culture chambers at a density of 2 × 10^3^ cells/cm^2^ and incubated for 6 days. The culture medium was replaced at day 11 and day 13. At day 14, the cells were harvested as on day 8. The cell suspension was placed in a transfer bag and washed with PBS. The cells were then resuspended in the final vehicle medium.	NA	MEM supp. with human platelet growth factor enriched plasma, 10 mg/mL ciprofloxa-cin and 1 U/mL heparin	NS	1	≥90%
25	Soler et al. [54]	BM-MSC	Fresh	120 mL of bone marrow was extracted from the posterior superior iliac spine.	Bone marrow cells were separated using an automated Sepax device and Ficoll–Paque reagent. MNCs were collected, washed and resuspended in culture medium.	Cells were plated at 2 × 10^5^ cells/cm^2^ onto cell culture plates and placed in a humidified 37 °C incubator with 5% CO_2_. The medium was changed every three to four days.	Trypsinization was performed when confluency was reached. Then, cells were replated at 1000 cells/cm^2^. On day 21, cells were harvested and washed with a saline solution and resuspended in the final vehicle medium.	NA	DMEM-low glucose supp. with 10% human serum.	70–90%	NS	85%
26	Lamo-Espin-osa et al. [55,56]	BM-MSC	Fresh	100 mLof bone marrow was extracted from the posterior superior iliac spine.	The MNC fraction was isolated by Ficoll density gradient centrifugation.	Cells, ranging between 20 × 10^6^ and 60 × 10^6^, were subsequently seeded in culture flasks with culture medium. Cells were incubated in a humidified 37 °C incubator with 5% CO_2_. The culture medium was changed every 3–4 days. About 10–15 days later, colonies were formed and the cells were split with TrypLE Select™ and seeded at 3000–5000 cells/cm^2^.	Once confluency was reached, cells were split again and cultured until they were available at the amounts required to be administered to patients. Finally, cells were harvested with TrypLE Select™, washed three times with PBS and resuspended in the final vehicle medium.	NA	Alpha-MEM without ribonucleo-sides supp. with 5% PL, 2 units/mL heparin, pen/strep at 1 % and 1 ng/mL bFGF	70–80%	NS	NS
27	Vega et al. [57]	BM-MSC	Fresh	Bone marrow samples (103 ± 8 mL) were harvested from the pelvic bone (iliac crest).	The MNC fraction was isolated by density gradient centrifugation.	Isolated cells were resuspended and cultured in culture medium and incubated in a humidified 37 °C incubator with 5% CO_2_. Periodic washing was carried out to remove non-adherent cells.	When cells reached confluency, they were trypsinized and replated, and the process was repeated for 2 more passages. At the end of this period (21–24 days), cells were harvested and resuspended in the final vehicle medium.	NA	DMEM-Low glucose, 10% FBS and 1% pen/strep	80%	2	>98%
28	Jo et al. [58,59]	AD-MSC	Not explic-itly stated possi-bly fresh.	Adipose tissue was obtained by lipoaspiration from the abdominal subcutaneous fat.	Adipose tissues were digested with collagenase I (1 mg/mL) under gentle agitation for 60 min at 37 °C. The digested tissues were filtered through a 100-um nylon sieve to remove cellular debris and were centrifuged at 470× *g* for 5 min. The pellet was resuspended in DMEM–based media containing 0.2 mM ascorbic acid and 10% FBS. The cell suspension was recentrifuged at 470× *g* for 5 min. The supernatant was discarded, and the cell pellet was collected.	The cell fraction was cultured overnight in DMEM-based media containing 0.2 mM ascorbic acid and 10% FBS. Cells were incubated in a humidified 37 °C incubator with 5% CO_2_. After 24 h, non-adherent cells were removed by washing with PBS. The cell medium was changed to culture medium. The cells were maintained for 4–5 days until confluent (P0). When the cells reached confluency, they were subculture-expanded until P3.	NS	NA	Keratinocyte-SFM based media containing 0.2 mM ascorbic acid, 0.09 mM calcium, 5 ng/mL rEGF, and 5% FBS	90%	3	>95%
29	Orozco et al. [60,61]	BM-MSC	Fresh	Bone marrow samples (103 ± 8 mL) were harvested from the pelvic bone (iliac crest).	The MNC fraction was isolated by density gradient centrifugation.	Isolated cells were resuspended and cultured in culture medium and incubated in a humidified 37 °C incubator with 5% CO_2_. Periodic washing was carried out to remove non-adherent cells.	When cells reached confluency, they were trypsinized and replated, and the process was repeated for 2 more passages. At the end of this period (21–24 days), cells were harvested and resuspended in the final vehicle medium.	NA	DMEM-Low glucose, 10% FBS and 1% pen/strep	80%	3	91% ± 6%
30	Emadedin et al. [62]	BM-MSC	Fresh	≈50 mL of bone marrow was extracted from the posterior superior iliac spine.	Bone marrow aspirate was added to 50 mL PBS and centrifuged at 1500× *g* for 20 min. MNC was then washed with PBS.	Isolated cells were plated at a density of 1 × 10^6^ cells/cm^2^ in 15 mL culture medium. Seven days after culture initiation, non-adherent cells were removed by medium replacement and cells were expanded through subcultures until P2.	P2 cells were washed with PBS and trypsinized with trypsin/ EDTA (0.2%). Finally, the cells were resuspended in the final vehicle medium.	NA	Alpha-MEM supp. with 100 IU pen/strep and 10% hyclon bovine serum	NS	2	NS
31	Davatchi et al. [63,64]	BM-MSC	Fresh	30 mL of bone marrow was obtained from each patient.	Using the Ficoll hypaque density gradient, the MNC of bone marrow were separated.	Isolated cells were seeded with 1 × 10^6^ MNCs/mL for primary culture in 21 mL culture medium. Cells were incubated in a humidified 37 °C incubator with 5% CO_2_. Every 4 days cells were fed by complete medium replacement, until the confluence of fibroblast-like cells at the base of flasks. Thereafter the adherent cells were re-suspended using 0.025% trypsin and reseeded at 1 × 10^4^ cells/mL. When cells reached confluence by the end of first passage, they were incubated only with M199 medium for one more day.	Cells were detached with trypsinization (0.025% trypsin) and washed with normal saline supp. with 2% HSA three times, then resuspended in the final vehicle medium.	NA	DMEM with 10% of FBS	NS	NS	NS
32	Centeno et al. [65]	BM-MSC	Fresh/Cryo	20 mL of bone marrow was extracted from the posterior superior iliac spine.	The aspirate was centrifuged at 200× *g* for 4–6 min to separate nucleated cells. The nucleated cells were placed in a separate centrifuge tube and pelleted by centrifuging at 1000× *g* for 6 min. The pellet was washed once in PBS, and then re-suspended in culture medium.	Nucleated cells were seeded in a tissue culture flask at 1 × 10^6^ cells/cm^2^ and incubated in a humidified 37 °C incubator with 5–17% CO_2_. Culture medium was changed after 48–72 h, removing non-adherent cells. MSC colonies that developed after 6–12 days in culture were harvested with use of TrypLE Select. To expand the MSCs, cells were re-plated at a density of 6000–12,000 cells/cm^2^ in culture medium and grown to near confluence.	After MSCs had been sub-cultured to the P2-P7, they were harvested with TrypLE Select, washed, and suspended in the final vehicle medium.	Frequently, MSCs for re-injection were cryogenically preserved prior to clinical use; they were first grown to confluency, harvested as described, and suspended at a density of 1–3 × 10^6^ cells/mL in either 90–95% auto PL or in platelet poor plasma. Cells were frozen to −80 °C in a controlled rate freezing device, and 24 to 72 h later, they were transferred to a −150°C freezer or into liquid nitrogen storage.	DMEM with 10–20% PL, 5 ug/mL doxycycline, and 2 IU/mL heparin	80%	2–7	NS
33	Centeno et al. [66]	BM-MSC	Fresh	20 mL of bone marrow was extracted from the posterior superior iliac spine.	Whole marrow was centrifuged at 100× *g* for 4–6 min. The plasma was removed, placed in a separate tube, and centrifuged at 1000× *g* for 10 min to pellet the nucleated cell fraction. The nucleated cells were washed once in PBS, counted, and then re-suspended in culture medium.	Isolated cells were seeded at 1 × 10^6^ cells/cm^2^ in flask culture and incubated in a humidified 37 °C incubator with 5% CO_2_. The culture medium was changed after 3 days. MSC colonies developed 6–12 days after seeding. After growing to near confluence, the colonies were trypsinized over 2–3 min.	Before reaching confluence, cells were harvested with 1 × trypsin in DPBS with 1 mM EDTA and the MSCs were reseeded at a density of 12,000 cells/cm^2^ in Alpha-MEM + 5%, 10%, or 20% PL. After MSCs had been grown to the P5, they were suspended in the final vehicle medium.	NA	DMEM with 10% PL	NS	5	NS

AD-MSC = Adipose-derived mesenchymal stromal cell, Alpha-MEM = Alpha-Minimum essential medium, Allo = Allogeneic, Auto = Autologous, bFGF = Basic fibroblast growth factor, BM-MSC = Bone marrow-derived mesenchymal stromal cell, Cryo = Cryopreserved, DMEM = Dulbecco’s Modified Eagle Medium, DMSO = Dimethyl sulfoxide, DPBS = Dulbecco’s Phosphate-Buffered Saline, EDTA = Ethylenediaminetetraacetic acid, FBS = Fetal bovine serum, Keratinocyte-SFM-based media = Keratinocyte-serum free medium, HSA = Human serum albumin KO-DMEM = Knockout Dulbecco’s Modified Eagle’s Medium, MEM = Minimum Essential Medium Eagle, MNC = Mononuclear cell, MSC = Mesenchymal stromal cell, NA = Not applicable, NS = Not specified, Pen/strep = Penicillin/streptomycin, PL = Platelet lysate, rEGF = Recombinant epidermal growth factor, RPM = Rounds per minute, Supp. = Supplemented, SVF = Stromal vascular fraction, UC-MSC = Umbilical cord-derived mesenchymal stromal cell.

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
