# Peer review of "A Narrative Review on Manufacturing Methods Employed in the Production of Mesenchymal Stromal Cells for Knee Osteoarthritis Therapy"

_biomedicines, 2025, doi:10.3390/biomedicines13020509_

Round 1

Reviewer 1 Report

Comments and Suggestions for Authors

This is a useful and comprehensive review of the current state of the rather disparate reporting techniques in the world of mesenchymal stromal cell injections. It highlights the lack of standardisation and therefore the inability of studies to come to meaningful comparative conclusions. They also propose a method of standardisation with adoption of minimal reporting criteria. This is helpful and will aid clinicians and comparison of studies in the future.

Author Response

Comment 1: This is a useful and comprehensive review of the current state of the rather disparate reporting techniques in the world of mesenchymal stromal cell injections. It highlights the lack of standardisation and therefore the inability of studies to come to meaningful comparative conclusions. They also propose a method of standardisation with adoption of minimal reporting criteria. This is helpful and will aid clinicians and comparison of studies in the future.

Response 1: We are happy to hear that the reviewer has found the review comprehensive and the minimal reporting criteria as a helpful initiative. We have not made any corrections to the manuscript based on reviewers 1 comment.

Of further note all changes in the manuscript text have been highlighted (1) red: requested by reviewer 2 and (2) green: requested by reviewer 3. Please find these changes highlighted in the manuscript version with tracked changes.

Thank you for your time and assistance.

If any other comments arise, I am happy to provide a quick response.

Sincerely,

Rasmus Roost Aabling, MD

Department of Clinical Medicine - Comparative Medicine Lab and SDCA-Steno Diabetes Center Aarhus

Aarhus University

Palle Juul-Jensens Boulevard 99 and 11

DK-8200, Aarhus N, Denmark

Email: rasaab@clin.au.dk

Reviewer 2 Report

Comments and Suggestions for Authors

The review delves into the variability of protocols used in MSC production and the challenges of comparing outcomes across studies. The manuscript is well-structured. There are some issues that need further clarification.

1. The safety and efficacy of MSC-based therapies are paramount. It is recommended that the manuscript includes a detailed discussion on the quality control measures implemented during the MSC manufacturing process.

2. The use of proteins and serum in vehicle medium can be avoided cell loss and poor viability. However, the use of proteins and serum can increase the risk of adverse reactions, especially when allogeneic serum is added. It is recommended that the vehicle medium section in minimal report explicitly indicates whether serum has been added, and if so, whether it is autologous or allogeneic.

Author Response

Comment 1: The review delves into the variability of protocols used in MSC production and the challenges of comparing outcomes across studies. The manuscript is well-structured. There are some issues that need further clarification.

  1. The safety and efficacy of MSC-based therapies are paramount. It is recommended that the manuscript includes a detailed discussion on the quality control measures implemented during the MSC manufacturing process.
  2. The use of proteins and serum in vehicle medium can be avoided cell loss and poor viability. However, the use of proteins and serum can increase the risk of adverse reactions, especially when allogeneic serum is added. It is recommended that the vehicle medium section in minimal report explicitly indicates whether serum has been added, and if so, whether it is autologous or allogeneic.

Response 1: We are deeply grateful for the insightful observations and opinions shared by the reviewer. We agree with these comments. Therefore, we have as answer to 1 about quality control reviewed the included clinical trials again accordingly.  

This is an important point, and we agree with the reviewer that it would be valuable in the manuscript to have additional focus on the quality control measures employed. We have therefore added:

  • additional details regarding quality control in a Table 4: Quality control measures during MSC manufacturing (Page 46 - 53); some of the aspects considered in Table 4 were previously in Table 2 (“Manufacturing process of MSCs in the clinical trials”) and these have thus now been removed from Table 2. 
  • a subsection focused on Quality control measures (Page 54 – 56 paragraph “Quality control measures” line 456 - 563).
  • a section on quality control to Table 5: Minimal reporting criteria for future clinical trials (Page 57).
  • As we have added a new table to the manuscript this has also been added to the start of the manuscript where the different tables are presented (Page 4, paragraph “Manufacturing of MSCs”, line 159-161).

As answer to 2 about clarification of the minimal reporting criteria for future clinical trials about serum (autologous or allogeneic) addition we have:

We agree with the reviewer’s recommendation and have now revised Table 5 “Minimal reporting criteria for future clinical trials” accordingly. We have now also further clarified and highlighted the importance of reporting the addition of serum (autologous or allogeneic) (Page 56, paragraph “Pre-injection Reconstitution”).

Of further note all changes in the manuscript text have been highlighted (1) red: requested by reviewer 2 and (2) green: requested by reviewer 3. Please find these changes highlighted in the manuscript version with tracked changes. Thank you for your time and assistance.

If any other comments arise, I am happy to provide a quick response.

Sincerely,

Rasmus Roost Aabling, MD

Department of Clinical Medicine - Comparative Medicine Lab and SDCA-Steno Diabetes Center Aarhus

Aarhus University

Palle Juul-Jensens Boulevard 99 and 11

DK-8200, Aarhus N, Denmark

Email: rasaab@clin.au.dk

Reviewer 3 Report

Comments and Suggestions for Authors

In this article, the authors have reviewed the literature on methods used to obtain, isolate, culture, harvest, identify, preserve, and administer mesenchymal cells used for reconstitution in patients with osteoarthritis. The article is difficult to follow as written and needs to be shortened and better organized. The reviewer suggests using headings (e.g., MSC sources, procurement, isolation,  culturing, etc.) rather than Tables to summarize their findings. The summary presented at the end of the article (Cellular Product Characteristics and Culture Process) can serve as a useful heading guideline. References should be annotated as a number with details included only in a Reference List at the end of the article. In summary, the article requires considerable revision in order to be accepted for publication.

Author Response

Comment 1: In this article, the authors have reviewed the literature on methods used to obtain, isolate, culture, harvest, identify, preserve, and administer mesenchymal cells used for reconstitution in patients with osteoarthritis. The article is difficult to follow as written and needs to be shortened and better organized. The reviewer suggests using headings (e.g., MSC sources, procurement, isolation,  culturing, etc.) rather than Tables to summarize their findings. The summary presented at the end of the article (Cellular Product Characteristics and Culture Process) can serve as a useful heading guideline. References should be annotated as a number with details included only in a Reference List at the end of the article. In summary, the article requires considerable revision in order to be accepted for publication.

Response 1: We appreciate the reviewer’s valuable feedback and have thoroughly edited the manuscript, in particular with a focus on shortening it and making it easier to read. We have additionally tried to build on the headings employed to further structure the manuscript. Please refer to manuscript with track changes for an overview of changes. As the Tables included in the manuscript present considerable additional data not contained in the text, and as these provide a quicker overview than presenting and discussing the data in paragraph format only, we have chosen to keep the Tables in their current format. Please note we have also added an additional discussion of quality control measures and a corresponding Table as suggested by Reviewer 2. 

Of further note all changes in the manuscript text have been highlighted (1) red: requested by reviewer 2 and (2) green: requested by reviewer 3. Please find these changes highlighted in the manuscript version with tracked changes.  

Thank you for your time and assistance.

If any other comments arise, I am happy to provide a quick response.

Sincerely,

Rasmus Roost Aabling, MD

Department of Clinical Medicine - Comparative Medicine Lab and SDCA-Steno Diabetes Center Aarhus

Aarhus University

Palle Juul-Jensens Boulevard 99 and 11

DK-8200, Aarhus N, Denmark

Email: rasaab@clin.au.dk

Round 2

Reviewer 3 Report

Comments and Suggestions for Authors

The comments made in my first review remain unchanged. The article needs to be more concise, with headings for each component involved in MSC reconstitution (see summary at end of article). I see little change in compliance with my original suggestions, and do not consider the article suitable for publication in biomedicines. 

Author Response

Comment:

The comments made in my first review remain unchanged. The article needs to be more concise, with headings for each component involved in MSC reconstitution (see summary at end of article). I see little change in compliance with my original suggestions, and do not consider the article suitable for publication in biomedicines. 

Response:

We thank the reviewer for their feedback, and would like to comply with these suggestions, however we are uncertain of the changes requested, and hope that the reviewer can elaborate further, perhaps by giving a few concrete examples.

We further apologize if we have misunderstood the reviewer’s initial comments. We believed that we had addressed the requested changes by rephrasing and shortening the text, and by implementing additional headings from “Table 5: Minimal reporting criteria for future clinical trials”. These headings have been added in several sections of the manuscript to improve its structure. Some examples are:

  • The heading “Cellular product characteristics and culture process” has been added (Page 30, line 5)
  • The heading “Pre-injection reconstitution” has been added (Page 42, line 258)
  • The heading “Injection Procedure” has been added (Page 44, line 409)
  • The heading “Quality control measures” has been added (Page 54, line 461)

In the first round the reviewer also requested not to use tables. As the tables included in the manuscript present considerable additional data not contained in the text, and as these provide a concise and accessible overview than presenting and discussing the data in paragraph format only, we have chosen to keep the tables in their current format.

Please note we have also added an additional discussion of quality control measures and a corresponding Table as suggested by Reviewer 2. 

Of further note all changes in the manuscript text have been highlighted (1) red: requested by reviewer 2 and (2) green: requested by reviewer 3. Please find these changes highlighted in the manuscript version with tracked changes.  

We hope you will be able to provide us with additional guidance in improving the manuscript and thank you for your time and assistance.

Sincerely,

Rasmus Roost Aabling, MD

Department of Clinical Medicine - Comparative Medicine Lab and SDCA-Steno Diabetes Center Aarhus

Aarhus University

Palle Juul-Jensens Boulevard 99 and 11

DK-8200, Aarhus N, Denmark

Email: rasaab@clin.au.dk

Round 3

Reviewer 3 Report

Comments and Suggestions for Authors

The authors have made some effort to shorten and improve the organization of the article. However, these efforts are insufficient to publish the article in its present form.

Comments on the Quality of English Language

Adequate

Author Response

Comment:

The authors have made some effort to shorten and improve the organization of the article. However, these efforts are insufficient to publish the article in its present form.

Response:

We thank the reviewer for the feedback and are pleased with the recognition of the changes that have been made. We apologize for the continued confusion and possible misunderstanding of the reviewer’s comment. As previously noted we believe tables serve an important function in summarizing the details of the included literature for an overall overview.

We thank you for your time and assistance.

Sincerely,

Rasmus Roost Aabling, MD

Department of Clinical Medicine - Comparative Medicine Lab and SDCA-Steno Diabetes Center Aarhus

Aarhus University

Palle Juul-Jensens Boulevard 99 and 11

DK-8200, Aarhus N, Denmark

Email: rasaab@clin.au.dk